# Insight into binding of endogenous neurosteroid ligands to the sigma-1 receptor

Chunting Fu[1,2], Yang Xiao[1,2], Xiaoming Zhou ®[1] ✉ & Ziyi Sun ®[1] ✉

The sigma-1 receptor (σ1R) is a non-opioid membrane receptor, which responds to a diverse array of synthetic ligands to exert various pharmacological effects. Meanwhile, candidates for endogenous ligands of σ1R have also been identified. However, how endogenous ligands bind to σ1R remains unknown. Here, we present crystal structures of σ1R from *Xenopus laevis* (xlσ1R) bound to two endogenous neurosteroid ligands, progesterone (a putative antagonist) and dehydroepiandrosterone sulfate (DHEAS) (a putative agonist), at 2.15-3.09 Å resolutions. Both neurosteroids bind to a similar location in xlσ1R mainly through hydrophobic interactions, but surprisingly, with opposite binding orientations. DHEAS also forms hydrogen bonds with xlσ1R, whereas progesterone interacts indirectly with the receptor through water molecules near the binding site. Binding analyses are consistent with the xlσ1R-neurosteroid complex structures. Furthermore, molecular dynamics simulations and structural data reveal a potential water entry pathway. Our results provide insight into binding of two endogenous neurosteroid ligands to σ1R.

The sigma-1 receptor (σ1R) is an endoplasmic reticulum (ER)-localized, non-opioid transmembrane receptor implicated in many pathological conditions, including neurodegenerative disorders and cancer[1]. It was initially thought to be an opioid receptor subtype that mediates the psychotomimetic effects of a prototypic benzomorphan ligand, *N*-allylnormetazocine (SKF-10,047)[2]. Cloning of σ1R reveals that σ1R is completely distinct from the classical μ-, κ-, and δ-opioid receptors, which are G protein-coupled receptors (GPCRs)[3]. A structurally diverse array of synthetic ligands of σ1R have been developed and studied[4,5], including both agonists (e.g. SKF-10,047, pentazocine, and PRE-084) and antagonists (e.g. haloperidol, NE-100, and S1RA). These synthetic ligands of σ1R exert various pharmacological effects, such as anti-amnesic, antipsychotic or antitumor effects[5]. Some of them (e.g. SA4503 and ANAVEX2-73) have entered clinical trials for treating affective or cognitive disorders[6]. Meanwhile, pharmacophore models have been proposed to describe the features of synthetic σ1R ligands[7], which usually contain a basic amine site that forms hydrogen bonds or salt bridges with a highly conserved glutamate residue of σ1R (E172 in human[8,9] and E169 in *Xenopus laevis*[10]).

Recently, structures of human σ1R (hσ1R) and *Xenopus laevis* σ1R (xlσ1R) bound to various synthetic ligands have been resolved (Supplementary Table 1), elucidating their binding mechanisms[8–10]. So far, all reported σ1R structures are homotrimers, with the ligand binding site located inside the β-barrel domain of each σ1R monomer[8–10] (Fig. 1a). The internal space (lumen) of the β-barrel domain adopts an elongated shape, which is sufficiently large to accommodate structurally diverse ligands (Supplementary Table 1). The residues lining the β-barrel lumen are mostly hydrophobic, except for a few hydrophilic patches in the distal space of the β-barrel lumen (away from the membrane)[8–10]. The highly conserved E172/E169 of σ1R is located within one of the hydrophilic patches. Consistently, all σ1R structures with bound agonists (pentazocine and PRE-084) or antagonists (PD144418, haloperidol, NE-100, and S1RA) demonstrate that the ligands are mainly surrounded by hydrophobic residues of σ1R, in contact with the ligands' hydrophobic moieties[8–10]. Furthermore, all these synthetic ligands contain a positively charged amine site that interacts directly with E172/E169 of σ1R (Supplementary Table 1), consistent with the pharmacophore models[7].

[1]Department of Integrated Traditional Chinese and Western Medicine, State Key Laboratory of Biotherapy, West China Hospital, Sichuan University, Chengdu, Sichuan, China. [2]These authors contributed equally: Chunting Fu, Yang Xiao. ✉e-mail: x.zhou@scu.edu.cn; ziyi.sun@scu.edu.cn

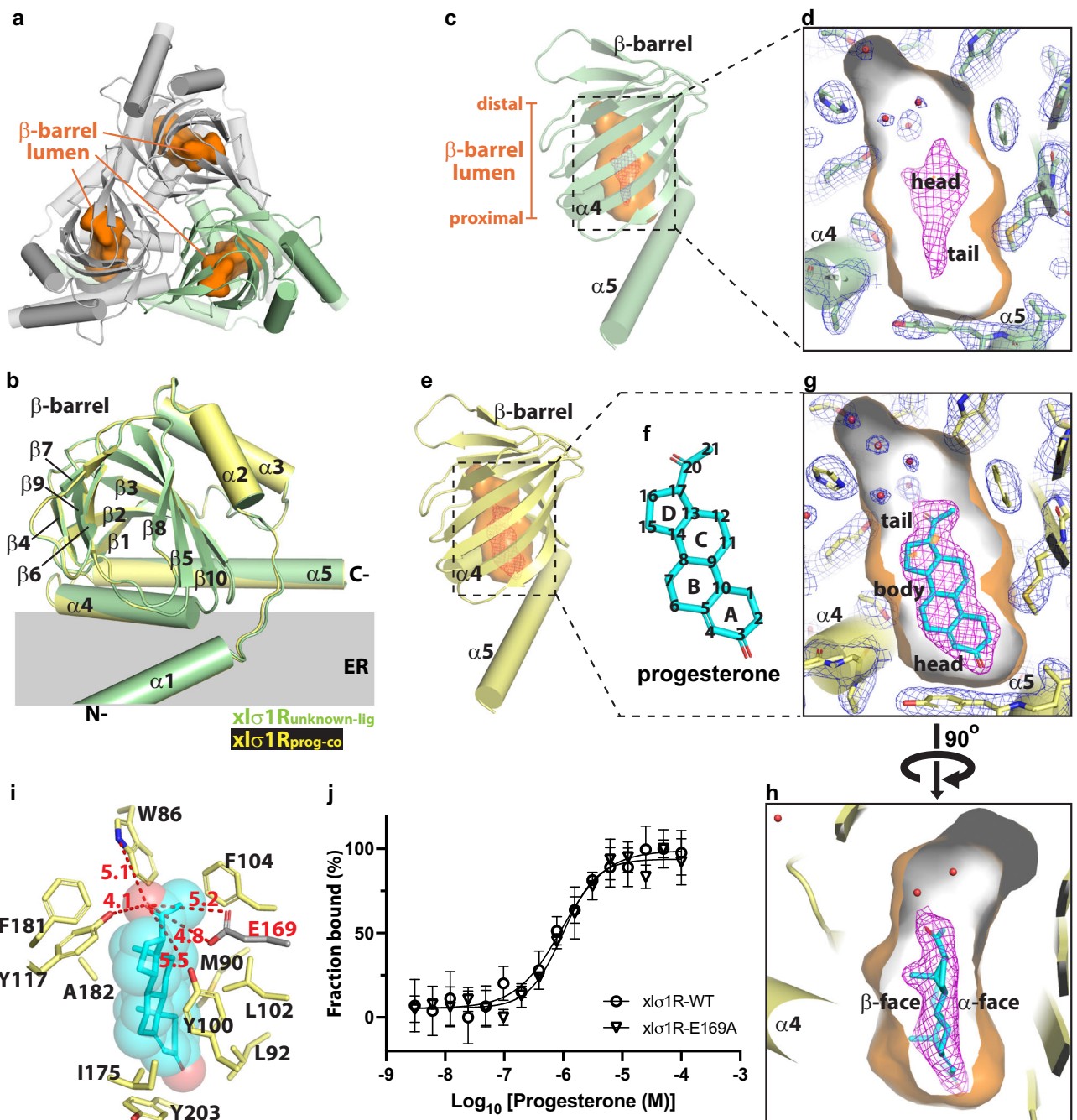

**Fig. 1 | Structure of xlσ1R bound to a putative antagonist progesterone.**
**a** Structure of the xlσ1R_unknown-lig homotrimer viewed perpendicular to the membrane from the β-barrel side. One protomer is colored in green and the other two in gray. The β-barrel lumen is rendered in orange surface throughout the manuscript. **b** Superposition of xlσ1R_unknown-lig (green cartoon) and xlσ1R_prog-co (yellow cartoon). Five α helices (α1-α5) and the β-barrel (β1-β10) are labeled. The relative position of the ER membrane is indicated by a gray rectangle. **c** An unknown electron density (purple mesh) inside the β-barrel lumen (orange surface) of xlσ1R_unknown-lig (green cartoon). The words "distal" and "proximal" describe the relative distances from different β-barrel lumen regions to the ER membrane. **d** A close-up view of the tadpole-shaped unknown density (purple mesh) from panel **c**. **e** A xlσ1R_prog-co protomer (yellow cartoon) containing a fish-shaped density (purple mesh) in the β-barrel lumen (orange surface). **f** Chemical structure diagram of progesterone. The diagram shows the α-face of progesterone's steroid rings A to D.

The backbone carbon atoms are numbered. **g**, **h** Fitting of progesterone (cyan sticks) into the fish-shaped density (purple mesh) of xlσ1R_prog-co (in yellow). **i** Hydrophobic interactions between bound progesterone (in cyan) and residues of xlσ1R_prog-co (yellow sticks). Distances from progesterone's C20 carbonyl oxygen to nearby oxygen- or nitrogen-containing side chains are indicated by red dashed lines with distances labeled. E169 is shown as gray sticks. **j** MST fitting curves of progesterone binding to xlσ1R wild-type (WT, open circles) and E169A (open triangles). MST measurements were repeated three times ($N = 3$ biologically independent samples) with similar results. Data are presented as mean ± SD. Source data are provided as a Source Data file. In panels **d**, **g**, and **h**, water molecules are rendered as red spheres. In panels **c**, **d**, **e**, **g**, and **h**, the purple mesh shows the simulated annealing $F_o$-$F_c$ (omit) map contoured at 3.0 σ level. In panels **d** and **g**, the blue mesh shows the simulated annealing $2F_o$-$F_c$ map contoured at 1.2 σ level. In panels **c** and **e**, only the β-barrel and α4/α5 of xlσ1R are shown for clearer views.

In the meantime, endogenous ligands of σ1R are less well-defined[11]. The search for endogenous σ1R ligands has yielded several candidates including neurosteroids[12,13], sphingolipids[14], *N,N*-dimethyltryptamine[15], myristic acid[16], and choline[17], which have been shown to bind to σ1R or modulate its activity. Among these candidates, neurosteroids were the first shown to bind to σ1R by in vitro and in vivo binding assays[12,18,19]. Neurosteroids are steroids synthesized in the brain that affect neuronal functions, and usually include progesterone, pregnenolone, pregnenolone sulfate, allopregnanolone, dehydroepiandrosterone (DHEA) and dehydroepiandrosterone sulfate (DHEAS)[19] (Supplementary Table 2). Among the neurosteroids tested, progesterone was the most potent inhibitor of radioligand (e.g. [³H]SKF-10,047) binding to σ1R, with $K_i$ values in the sub-micromolar range[12,13]. Other neurosteroids, such as pregnenolone sulfate, DHEA and DHEAS, showed weaker potencies with $K_i$ values in the micromolar range or more[12,13]. Moreover, physiological tests also demonstrate the interactions between neurosteroids and σ1R in vivo. For instance, the *N*-methyl-D-aspartate (NMDA)-evoked [³H]norepinephrine release from preloaded rat hippocampal slices was potentiated by DHEAS and inhibited by pregnenolone sulfate[20]. These effects were blocked by σ1R antagonists haloperidol and BD1063, and by progesterone[20]. In addition, DHEA was shown to potentiate the excitatory response of pyramidal neurons to NMDA in the CA₃ region of rat dorsal hippocampus, which was blocked by σ1R antagonists haloperidol and NE-100, and also by progesterone[21]. On the other hand, amnesia and learning impairments induced in animal models could be attenuated by σ1R agonists such as SKF-10,047, pentazocine, PRE-084, and SA4503[22–24], as well as by neurosteroids like pregnenolone sulfate, DHEA and DHEAS[25,26]. These beneficial effects were blocked or antagonized by σ1R antagonists such as BMY-14,802, NE-100 and haloperidol, and by progesterone[22,25–27]. Therefore, progesterone is generally considered a putative σ1R antagonist, while the other neurosteroids are often considered putative σ1R agonists[19].

However, how endogenous ligands (including neurosteroids) bind to σ1R remains unknown. Recent computational studies have generated docking models for several endogenous ligand candidates of σ1R[28,29], providing valuable information on this subject. Nevertheless, lack of experimental structures of σ1R bound to endogenous ligands hinders our understanding towards the endogenous function and working mechanism of σ1R.

In this study, we report crystal structures of xσ1R bound to two neurosteroids, a putative antagonist progesterone and a putative agonist DHEAS. Combined with binding assays and molecular dynamics simulations, our results provide insight into the binding mechanism of two endogenous neurosteroid ligands to σ1R.

## Results

### Improving diffraction of xσ1R crystals for clear ligand identification

Previously, we have determined crystal structures of σ1R from *Xenopus laevis* (xσ1R) in complex with two synthetic ligands, PRE-084 and S1RA (Supplementary Table 1), by soaking the ligands into xσ1R crystals[10]. Therefore, to obtain crystals of xσ1R bound to neurosteroids, our first strategy was to soak various neurosteroids into xσ1R crystals. However, identification of the bound ligand in a xσ1R structure is challenging due to two reasons. First, the existing crystallization conditions for xσ1R only yield crystals with mediocre diffracting abilities (3–4 Å)[10], and the process of soaking usually decreases the diffraction quality of xσ1R crystals. Second, xσ1R structures contain an unidentified electron density in the ligand binding site even if no ligand was added during crystallization[10], which complicates the identification of the bound molecule, especially at low resolutions (e.g., 3.5–4.0 Å). Indeed, existing xσ1R crystals diffracted to ~4.0 Å after being soaked with neurosteroids, making it difficult to tell if the

electron density in the ligand binding site is the soaked neurosteroid or not.

To improve the diffraction of xσ1R crystals for clear identification of bound ligands, xσ1R was re-screened for better crystallization conditions. In the end, xσ1R with the purification tag removed by the tobacco etch virus (TEV) protease produced crystals of much higher quality than previously reported (3.0–3.5 Å)[10], and its structure was solved to 2.17 Å (termed xσ1R_unknown-lig; Table 1 and Supplementary Fig. 1). The xσ1R_unknown-lig crystal was packed in the I432 space group, with the asymmetric unit containing one xσ1R protomer (Supplementary Fig. 2a). Three xσ1R monomers assemble into a homotrimer in xσ1R_unknown-lig (Fig. 1a and Supplementary Fig. 2b), similar to previously reported σ1R structures[8–10]. The xσ1R_unknown-lig monomer contains five α-helices (α1 to α5) and a β-barrel domain that consists of ten β-strands (β1 to β10; Fig. 1b). The amino-terminal α1 is the only transmembrane helix, followed by two connecting helices α2 and α3. The ligand binding site is located in the lumen of the β-barrel domain (β1-β10), whose opening is covered by two membrane-adjacent helices (α4/α5) at the carboxy-terminus (Fig. 1b). Notably, in the xσ1R_unknown-lig structure (and the other xσ1R structures in this study), two xσ1R trimers packed in a membrane-side-to-membrane-side manner (Supplementary Fig. 2a), forcing α1 helices between the two trimers to fold inside towards the trimer bottom. As a result, α1 of xσ1R_unknown-lig (and the other xσ1R structures in this study) is more tilted than previously reported (Fig. 1b and Supplementary Fig. 2c). Since α1 does not participate directly in ligand binding inside the β-barrel lumen, α1 was not included during discussion of neurosteroid ligand binding in xσ1R in this study. Interestingly, cholesterol has been shown to interact with α1, which may regulate α1 orientation and play a role in σ1R oligomerization[30]. Unfortunately, no cholesterol (or cholesteryl hemisuccinate) molecule has co-crystallized with hσ1R or xσ1R to date, even though cholesterol (or cholesteryl hemisuccinate) was present during hσ1R/xσ1R crystallization[8–10].

Except for the flexible orientation of α1, superposition of the xσ1R_unknown-lig protomer onto previous xσ1R structures yielded all-atom root mean square deviations (RMSDs) of 0.3–0.4 Å (Supplementary Fig. 2c), indicating that the fold of xσ1R_unknown-lig is similar to the reported xσ1R structures. As seen before[10], though no known ligand was added during purification or crystallization, the xσ1R_unknown-lig structure contains an unidentified electron density in the β-barrel lumen, occupying the ligand binding site (Fig. 1c). The shape of the unidentified molecule resembles a tadpole with a bulky head and a slim tail (Fig. 1d). Although the identity of the unidentified molecule in xσ1R_unknown-lig was not resolved, the improved resolution (2.17 Å) offers more structural features of the unidentified density, allowing a clearer distinction between bound neurosteroids and the unidentified molecule.

### Structures of xσ1R in complex with a putative antagonist progesterone

Several neurosteroids, including progesterone, pregnenolone, pregnenolone sulfate, DHEA and DHEAS (Supplementary Table 2), were subjected to soaking to the xσ1R_unknown-lig crystals. Solvents used for solubilizing the neurosteroids were also screened to minimize their damage to the diffraction quality of xσ1R crystals. Eventually, xσ1R crystals soaked with progesterone, a putative σ1R antagonist, diffracted to ~2.5 Å and its structure was solved to 2.68 Å (termed xσ1R_prog-soak; Table 1 and Supplementary Fig. 1). Meanwhile, in addition to the soaking method, co-crystallization of xσ1R and progesterone was carried out using the crystallization condition of xσ1R_unknown-lig. Fortunately, co-crystallization of the xσ1R-progesterone complex produced higher-quality crystals that diffracted to ~2 Å and its structure was solved to 2.15 Å (termed xσ1R_prog-co; Table 1 and Supplementary Fig. 1). Since xσ1R_prog-soak and xσ1R_prog-co are nearly identical with an all-atom RMSD of ~0.1 Å (Supplementary

**Table 1 | Data collection and refinement statistics for the xlσ1R structures**

| PDB ID | xlσ1R$_{prog-co}$ 8W4B | xlσ1R$_{prog-soak}$ 8W4C | xlσ1R$_{unknown-lig}$ 8W4D | xlσ1R$_{DHEAS-I432}$ 8WWB | xlσ1R$_{DHEAS-C2}$ 8WUE | xlσ1R$_{side-open}$ 8W4E | xlσ1R$_{side-open-all}$ 8YBB |
|---|---|---|---|---|---|---|---|
| Data collection | | | | | | | |
| Space group | I 4 3 2 | I 4 3 2 | I 4 3 2 | I 4 3 2 | C 1 2 1 | C 1 2 1 | C 1 2 1 |
| Wavelength (Å) | 0.9785 | 0.9785 | 0.9785 | 0.9785 | 0.9785 | 0.9785 | 0.9786 |
| Unit cell | | | | | | | |
| a, b, c (Å) | 161.2, 161.2, 161.2 | 161.2, 161.2, 161.2 | 160.5, 160.5, 160.5 | 160.7, 160.7, 160.7 | 90.6, 144.7, 137.5 | 89.7, 145.7, 145.9 | 86.9, 146.8, 145.8 |
| α, β, γ (°) | 90, 90, 90 | 90, 90, 90 | 90, 90, 90 | 90, 90, 90 | 90, 104.2, 90 | 90, 107.4, 90 | 90, 107.0, 90 |
| Resolution (Å) | 2.15 (2.23–2.15) | 2.68 (2.78–2.68) | 2.17 (2.25–2.17) | 2.50 (2.59–2.50) | 3.09 (3.20–3.09) | 2.81 (2.91–2.81) | 3.12 (3.23–3.12) |
| Unique reflections | 18,962 (1938) | 10,389 (1016) | 18,902 (1843) | 12,544 (1246) | 31,523 (3160) | 43,727 (4356) | 31,055 (3078) |
| Multiplicity | 23.2 (20.0) | 6.6 (7.0) | 17.5 (17.4) | 9.7 (10.1) | 6.1 (6.2) | 6.7 (6.9) | 5.6 (5.8) |
| Completeness (%) | 96.2 (100.0) | 99.5 (100.0) | 99.9 (100.0) | 99.8 (100.0) | 99.8 (99.9) | 99.8 (99.9) | 99.8 (99.6) |
| I/σI | 16.0 (1.5) | 9.1 (1.6) | 17.2 (1.6) | 17.9 (1.7) | 15.0 (1.8) | 16.0 (1.7) | 15.7 (1.6) |
| $R_{merge}$ | 0.119 (2.038) | 0.148 (0.990) | 0.145 (1.837) | 0.100 (1.371) | 0.103 (1.050) | 0.072 (0.980) | 0.091 (1.101) |
| $R_{meas}$ | 0.122 (2.090) | 0.161 (1.069) | 0.149 (1.892) | 0.106 (1.444) | 0.113 (1.147) | 0.079 (1.060) | 0.100 (1.209) |
| $R_{pim}$ | 0.025 (0.464) | 0.061 (0.400) | 0.035 (0.452) | 0.033 (0.447) | 0.045 (0.455) | 0.030 (0.401) | 0.042 (0.494) |
| CC$_{1/2}$ | 0.998 (0.612) | 0.995 (0.673) | 0.999 (0.630) | 0.999 (0.632) | 0.998 (0.741) | 0.999 (0.769) | 0.999 (0.759) |
| Refinement | | | | | | | |
| Resolution (Å) | 2.15 (2.27–2.15) | 2.68 (2.95–2.68) | 2.17 (2.29–2.17) | 2.50 (2.70–2.50) | 3.09 (3.19–3.09) | 2.81 (2.87–2.81) | 3.12 (3.22–3.12) |
| No. reflections | 18,955 (1938) | 10,382 (1016) | 18,897 (1843) | 12,542 (1246) | 31,507 (3158) | 43,684 (4354) | 31,048 (3073) |
| Completeness (%) | 96.2 | 99.7 | 100.0 | 99.8 | 99.9 | 99.8 | 99.8 |
| $R_{work}$/$R_{free}$ (%) | 19.8/21.4 | 21.7/23.9 | 21.2/23.8 | 21.8/25.4 | 21.4/24.0 | 24.3/26.8 | 23.5/26.7 |
| No. atoms | 1867 | 1818 | 1858 | 1814 | 10,728 | 10,538 | 10,512 |
| Protein | 1769 | 1769 | 1769 | 1769 | 10,578 | 10,535 | 10,512 |
| Ligands | 23 | 23 | | 25 | 150 | | |
| Solvent | 75 | 26 | 89 | 20 | | 3 | |
| Average B-factor | 56.28 | 56.56 | 46.02 | 62.86 | 94.07 | 97.38 | 102.39 |
| Protein | 56.46 | 56.64 | 46.08 | 62.75 | 93.99 | 97.38 | 102.39 |
| Ligands | 53.36 | 61.58 | | 75.97 | 99.80 | | |
| Solvent | 52.99 | 46.84 | 44.84 | 56.08 | | 75.14 | |
| Ramachandran | | | | | | | |
| Favored (%) | 99.10 | 98.65 | 99.10 | 99.10 | 98.49 | 98.72 | 98.04 |
| Allowed (%) | 0.90 | 1.35 | 0.90 | 0.90 | 1.51 | 1.28 | 1.96 |
| Outliers (%) | 0.00 | 0.00 | 0.00 | 0.00 | 0.00 | 0.00 | 0.00 |
| RMS bonds (Å) | 0.004 | 0.004 | 0.003 | 0.002 | 0.003 | 0.001 | 0.002 |
| RMS angles (°) | 0.784 | 0.736 | 0.639 | 0.487 | 0.626 | 0.428 | 0.491 |
| Clashscore | 0.84 | 1.96 | 2.84 | 1.97 | 4.43 | 3.86 | 4.63 |

Statistics for the highest-resolution shell are shown in parentheses.

Fig. 2d), the higher-resolution xlσ1R$_{prog-co}$ structure was used for analysis of progesterone binding in xlσ1R.

The xlσ1R$_{prog-co}$ structure was solved in the I432 space group with one xlσ1R protomer per asymmetric unit, and three xlσ1R protomers form a homotrimer (Supplementary Fig. 2b). Comparison of the xlσ1R$_{prog-co}$ protomer to the xlσ1R$_{unknown-lig}$ protomer revealed that the two structures are highly similar with an all-atom RMSD of 0.19 Å (Fig. 1b). However, distinct from the tadpole-shaped density in xlσ1R$_{unknown-lig}$ (Fig. 1d), the electron density in the ligand binding site of xlσ1R$_{prog-co}$ resembles a fish with a small head, a wide body and a short tail (Fig. 1e, g). Interestingly, structural alignment of the lumen-lining residues between xlσ1R$_{prog-co}$ and xlσ1R$_{unknown-lig}$ yielded an all-atom RMSD of 0.22 Å, suggesting that the ligand binding residues of the two structures change very subtly when binding to the two different molecules (unknown ligand vs. progesterone; Supplementary Fig. 2e).

Meanwhile, the 2.15 Å-resolution structure of xlσ1R$_{prog-co}$ readily allowed a clear placement of a progesterone molecule into the fish-shaped density (Fig. 1f, g). The C3 carbonyl oxygen on A-ring of progesterone occupies the fish-shape's head proximal to the ER membrane, whereas the C17 keto methyl group on D-ring occupies the fish-shape's tail reaching towards the distal space of the β-barrel lumen (Fig. 1f, g). The steroid rings of progesterone become the fish-shape's body. Meanwhile, two density protrusions corresponding to the C10 and C13 methyl groups on the β-face of progesterone are clearly visible and point to the membrane (Fig. 1h).

The progesterone molecule binds inside the β-barrel domain to a location similar to where synthetic ligands (PRE-084 and S1RA) bind in previously reported xlσ1R structures[10]. However, unlike PRE-084 and S1RA, which contain a basic amine that interacts with the conserved E169 on the β10 strand in xlσ1R[10] (Supplementary Table 1), progesterone contains no amine group but two carbonyl (C3 and C20) oxygens that are capable of hydrogen bonding (Fig. 1f and Supplementary Table 2). Surprisingly, no oxygen, nitrogen or sulfur atom of xlσ1R is located within hydrogen bond distance[31] from the C3 or C20 carbonyl oxygen atom of progesterone, suggesting that no direct hydrogen bond is present between bound progesterone and the receptor (including E169) in xlσ1R$_{prog-co}$ (Fig. 1i). Consistently, mutation of E169

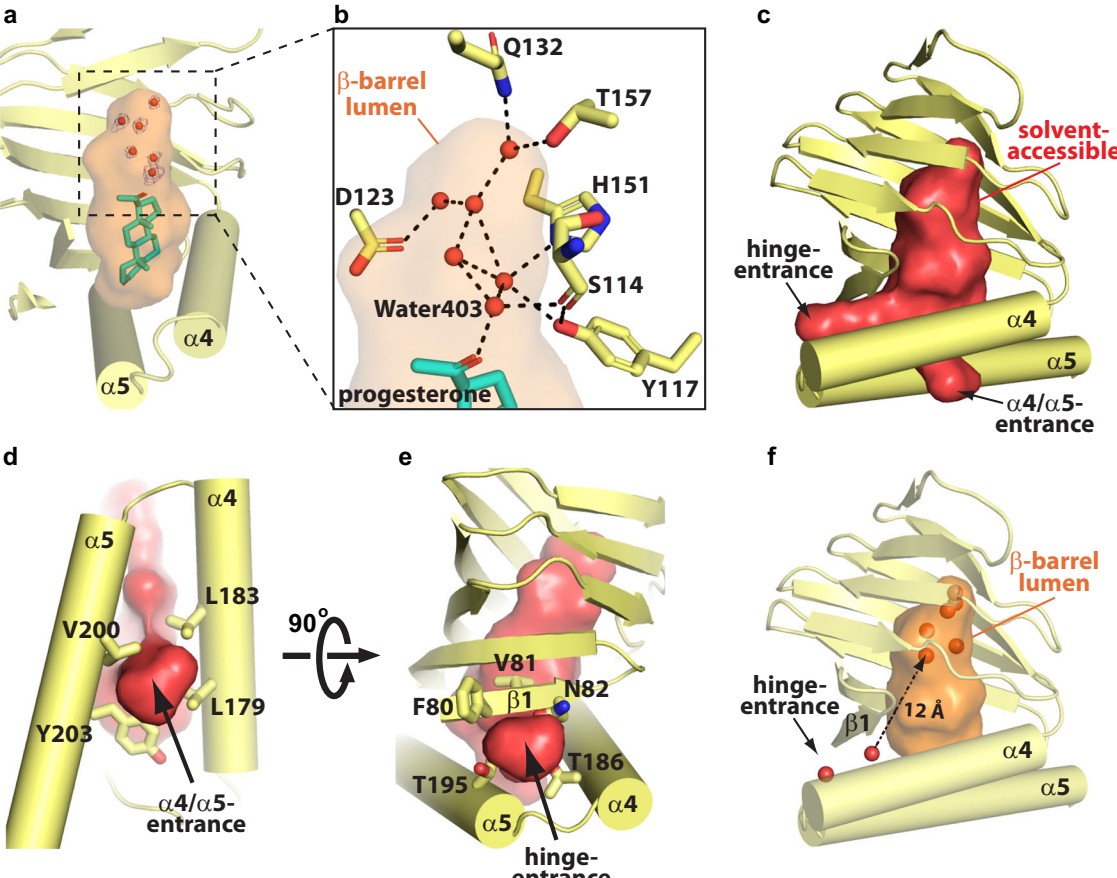

**Fig. 2 | Indirect interaction between progesterone and xlσ1R mediated by water. a** Electron densities (blue mesh) of six water molecules (red spheres) in the distal space of the β-barrel lumen (orange surface) of xlσ1R$_{prog-co}$ (yellow cartoon). The blue mesh shows the simulated annealing 2F$_o$-F$_c$ map contoured at 1.2 σ level for water molecules. Progesterone (cyan sticks) binds in the β-barrel lumen region proximal to the membrane. **b** A close-up view of panel **a** showing the distal space of the β-barrel lumen. Potential hydrogen bonds with the donor-acceptor distance of 2.7−3.6 Å are indicated by black dashed lines. Coordinating residues are shown as yellow sticks. Water403 is labeled. **c** Solvent accessibility analysis of xlσ1R$_{prog-co}$. The solvent-accessible space is displayed as red surface. The α4/α5-entrance and the hinge-entrance are indicated by black arrows. **d** The α4/α5-entrance surrounded by residues L179 (α4), L183 (α4), V200 (α5) and Y203 (α5) (yellow sticks). Viewed perpendicular to the membrane from the membrane side of xlσ1R$_{prog-co}$. **e** The hinge-entrance surrounded by residues F80 (β1), V81 (β1), N82 (β1), T186 (α4), and T195 (α5) (yellow sticks). **f** Two water molecules (red spheres) near the hinge-entrance in xlσ1R$_{prog-co}$. The shortest distance (-12 Å) between the hinge-entrance water and the distal β-barrel lumen water is indicated by a black dashed arrow. In panels **a**, **c**, **d–f**, only the β-barrel and α4/α5 of xlσ1R are shown for clearer views.

to alanine (E169A) in xlσ1R did not affect progesterone binding significantly, with an equilibrium dissociation constant $K_d = 0.94 \pm 0.22$ μM for the wild-type receptor and $K_d = 0.98 \pm 0.18$ μM for the mutant ($P = 0.78$; Fig. 1j and Supplementary Table 3). As a result, synthetic ligands, e.g. pentazocine, PD144418, NE-100, haloperidol, PRE-084 and S1RA, bind more deeply towards the distal end of the β-barrel lumen to interact directly with E172/E169 (β10) of σ1R (Supplementary Fig. 2f). Progesterone binds slightly closer towards the membrane (Supplementary Fig. 2f), mainly through hydrophobic interactions with W86 (β2), M90 (β2), L92 (β2), Y100 (β3), L102 (β3), F104 (β3), Y117 (β4/β5 loop), I175 (α4), F181 (α4), A182 (α4), and Y203 (α5) (Fig. 1i). These hydrophobic residues are highly conserved in σ1R among different species[10] (Supplementary Fig. 3), suggesting a conserved binding pattern for progesterone in σ1R. Furthermore, lack of a direct interaction between progesterone and E169/E172 (β10) of σ1R may account for the affinity difference between the synthetic ligands (usually in nM range)[4,32] and progesterone (in μM range)[12,13].

### Interaction between xlσ1R and progesterone mediated by water

Though the progesterone molecule in xlσ1R$_{prog-co}$ occupies only -2/3 of the β-barrel lumen space proximal to the ER membrane, the distal end of the β-barrel lumen is not empty (Fig. 2a). Interestingly, electron

densities corresponding to six water molecules (Water403, 404, 434, 447, 473, and 474) were observed within the distal space of the β-barrel lumen (Fig. 2a). In xlσ1R$_{prog-co}$, these water molecules form extensive hydrogen bonds with each other and with the hydrophilic patches in the distal region of the β-barrel lumen, including S114 (β4/β5 loop), Y117 (β4/β5 loop), D123 (β5), Q132 (β6), H151 (β8), and T157 (β9) (Fig. 2b). These hydrophilic residues are also conserved in different σ1R proteins[10] (Supplementary Fig. 3), suggesting a conserved function for these residues. Water403 also forms a direct hydrogen bond with the C20 carbonyl oxygen of the progesterone tail (Fig. 2b). Thus, this water-mediated hydrogen bond network connects bound progesterone indirectly to the receptor. Water densities were also observed in the β-barrel lumen of xlσ1R$_{unknown-lig}$ at similar locations (Fig. 1d), even though the unidentified ligand was not modeled. The involvement of water in the distal β-barrel lumen may help stabilize binding of progesterone (or other endogenous ligands) in σ1R, which has not been seen for synthetic ligands in previous σ1R structures[8–10]. Indeed, water molecules in protein-ligand binding sites are known to influence ligand binding, e.g. the specificity and affinity[33,34].

To provide a quantitative estimate of the role of the water molecules in the distal β-barrel lumen of xlσ1R$_{prog-co}$, the interaction/binding energy between xlσ1R and bound progesterone was calculated

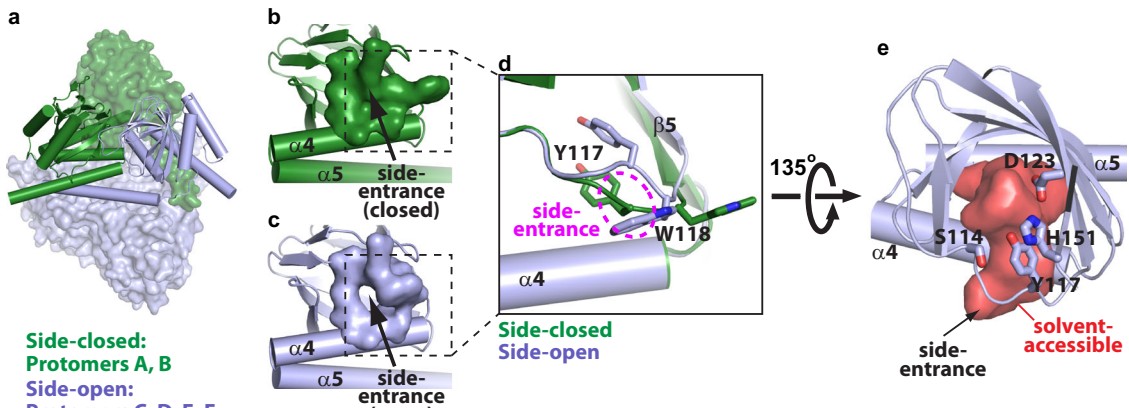

**Fig. 3 | A side-entrance for potential water entry in xlσ1R. a** Structure of xlσ1R_side-open with each asymmetric unit containing two trimers (six protomers). Protomers A and B are colored in green (side-closed conformation) and the rest in light blue (side-open conformation). **b, c** The side-closed conformation (**b**) and the side-open conformation (**c**). Residues 114-118 (β4/β5 loop) and 177-181 (α4) are displayed in surface mode. The location of the side-entrance is indicated by a black arrow. **d** Superposition of the side-open conformation (in light blue) onto the side-closed conformation (in green). Residues Y117 and W118 are displayed as sticks. The location of the side-entrance is indicated by a magenta dashed oval. **e** Solvent accessibility analysis of the side-open conformation viewed from the β-barrel side. The solvent-accessible space is shown as red surface, and residues capable of hydrogen bonding along the solvent path are shown as sticks. The side-entrance is indicated by a black arrow. In panels **b, c**, and **e**, only the β-barrel and α4/α5 of xlσ1R are shown for clearer views.

using the AMMOS2 web server[35] or BIOVIA Discovery Studio (see Methods). Consistently, the interaction energy between progesterone and xlσ1R decreased from −58.6 kcal/mol (no water bound), to −59.6 kcal/mol (Water403 bound), and to −62.0 kcal/mol (six water molecules bound) as calculated by AMMOS2 (Supplementary Table 4). Meanwhile, the binding energy between progesterone and xlσ1R decreased from −48.8 kcal/mol (no water bound) to −53.0 kcal/mol (six water molecules bound) as estimated by BIOVIA Discovery Studio (Supplementary Table 4). This data supports that the water molecules in the distal β-barrel lumen contribute to progesterone binding in xlσ1R.

## Potential water entry pathways

How does water enter the β-barrel lumen of σ1R? To find potential water entrances, solvent accessibility was analyzed in the xlσ1R_prog-co structure, and two paths were identified that connect the β-barrel lumen to the outside milieu (Fig. 2c). One entrance lies near L179 (α4), L183 (α4), V200 (α5), and Y203 (α5) between the α4 and α5 helices (termed α4/α5-entrance; Fig. 2d), which has been proposed for ligand entry by Meng and colleagues[10]. Since the α4/α5-entrance faces the ER membrane, it is less likely an entrance for water. The second entrance is located between the β1 strand (near F80, V81, and N82) and the hinge of the α4/α5 helices near T186 (α4) and T195 (α5) (termed hinge-entrance; Fig. 2e). Interestingly, two water molecules were observed near the hinge-entrance of xlσ1R_prog-co (Fig. 2f), supporting it as a potential entrance for water to enter the β-barrel lumen. A similar hinge-entrance with nearby water molecules has also been seen in the xlσ1R_unknown-lig structure (Supplementary Fig. 4a) and the human σ1R structure bound to (+)-pentazocine[9]. However, from the hinge-entrance to the distal lumen space where water was observed in xlσ1R_prog-co, the water molecules would have to travel at least 12 Å along a mostly hydrophobic path (Fig. 2f and Supplementary Fig. 4b). This data suggests that there may be other water entrance(s) on σ1R that allows water to reach the distal region of the β-barrel lumen more easily.

To find potential water entrances, three 100-ns molecular dynamics (MD) simulation runs were carried out in parallel, using a xlσ1R_prog-co monomer as an input model. Interestingly, in addition to the α4/α5-entrance and the hinge-entrance, the simulations revealed a third potential entrance located between the α4 helix (near S177 and

F181) and the β4/β5 loop near R116 (termed side-entrance; Supplementary Fig. 5a–d). The side-entrance appeared in all simulation trajectories and remained open for 30–60% of the simulation duration in different runs (Supplementary Fig. 5b–d). We thought that the side-entrance open conformation of xlσ1R may be stable enough to capture in crystal structures. Therefore, we solved structures for ~30 randomly selected xlσ1R crystals and eventually succeeded in capturing a xlσ1R structure at 2.81 Å with the side-entrance open (termed xlσ1R_side-open; Table 1 and Supplementary Fig. 1).

The xlσ1R_side-open structure was solved in the C2 space group, with six protomers (two trimers) in each asymmetric unit (Fig. 3a). Among them, two protomers (A and B) are in a side-closed conformation similar to xlσ1R_unknown-lig/xlσ1R_prog-co (Fig. 3b). The other four protomers (C, D, E and F) adopt a side-open conformation with an open side-entrance (Fig. 3c), similar to that observed in the simulations (Supplementary Fig. 5b-5d). Solvent accessibility analysis of the side-open protomer (e.g. protomer C) showed a clear opening surrounded by residues 114-118 (β4/β5 loop) and 177-181 (α4) (Fig. 3c). Comparison of the side-open and the side-closed conformations revealed that a local conformational rearrangement of Y117 and W118 on the β4/β5 loop leads to open or close of the side-entrance (Fig. 3d). Y117 and W118 are located near the trimer interface between two adjacent protomers within a trimer (Supplementary Fig. 6a). In xlσ1R_side-open, W118 of a side-open protomer (e.g. protomer C) is 4–5 Å away from Q191 and F193 (α4/α5 loop) of protomer A (Supplementary Fig. 6b). Meanwhile, Y117 or W118 of a side-closed protomer (e.g. protomer A) is not in proximity to its adjacent protomer B (Supplementary Fig. 6c). Therefore, there is sufficient space at the trimer interface for conformational change of Y117 and W118 between the side-closed and side-open conformations. Recently, we used the same strategy to solve ~30 additional xlσ1R structures and obtained a xlσ1R structure at 3.12 Å with the side-entrance open in all protomers (termed xlσ1R_side-open-all; Table 1 and Supplementary Fig. 1). Similar to xlσ1R_side-open, the xlσ1R_side-open-all structure was also solved in the C2 space group and contains two xlσ1R trimers in the asymmetric unit (Supplementary Fig. 6d). Differently, all six protomers of xlσ1R_side-open-all adopt the side-open conformation (Supplementary Fig. 6e), with all-atom RMSDs of 0.1–0.2 Å between protomers (Supplementary Fig. 6f). This data suggests that the conformational change of Y117 and W118 (β4/β5

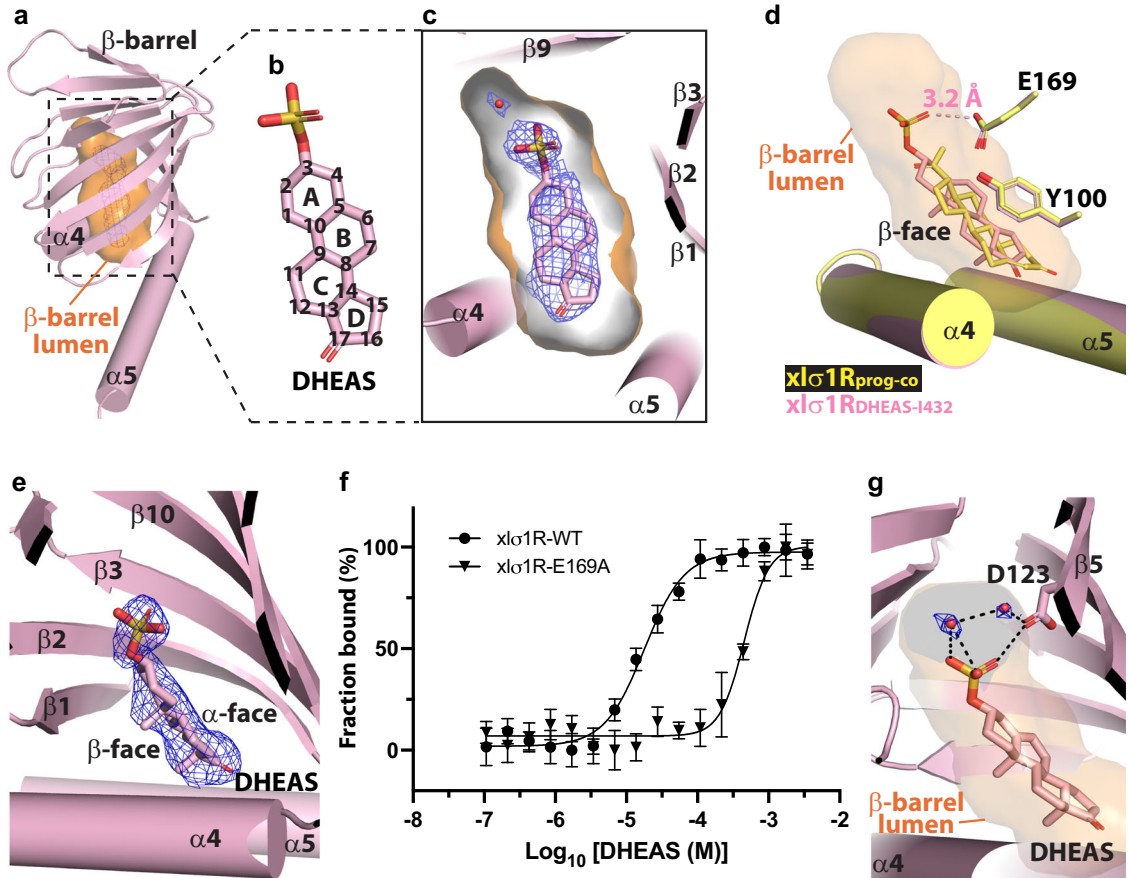

**Fig. 4 | Structure of xlσ1R bound to a putative agonist DHEAS. a** A xlσ1R$_{DHEAS-I432}$ protomer (pink cartoon) containing an electron density (blue mesh) in the β-barrel lumen (orange surface). For a clearer view, only the β-barrel and α4/α5 of xlσ1R are shown. **b** Chemical structure diagram of DHEAS. The diagram shows the α-face of the steroid rings A to D of DHEAS. The backbone carbon atoms are numbered. **c** Density fitting of DHEAS (pink sticks) in the β-barrel lumen (orange surface) of xlσ1R$_{DHEAS-I432}$ (pink cartoon). Water molecules in the distal β-barrel lumen are shown as red spheres. **d** Structural alignment of xlσ1R$_{prog-co}$ (in yellow) and xlσ1R$_{DHEAS-I432}$ (in pink). Progesterone, DHEAS and residues E169 and Y100 are shown as sticks. A potential hydrogen bond between the C3 sulfuric ester group of DHEAS and the E169 side chain is indicated by a pink dashed line with the donor-acceptor length labeled. Only α4/α5 of xlσ1R are shown for a clearer view. **e** The electron density (blue mesh) of DHEAS (pink sticks) in xlσ1R$_{DHEAS-I432}$, showing two density protrusions towards the membrane. **f** MST fitting curves of DHEAS binding to xlσ1R WT (closed circles) and E169A (closed triangles). MST measurements were repeated three times ($N = 3$ biologically independent samples) with similar results. Data are presented as mean ± SD. Source data are provided as a Source Data file. **g** Electron densities (blue mesh) of two water molecules (red spheres) in the distal β-barrel lumen (orange surface) of xlσ1R$_{DHEAS-I432}$. DHEAS and residue D123 are shown as sticks. Potential hydrogen bonds are indicated by black dashed lines. In panels **a, c, e,** and **g,** the blue mesh shows the simulated annealing 2F$_o$-F$_c$ map contoured at 1.2 σ level.

loop) between the side-closed and side-open conformations of xlσ1R is dynamic during crystallization.

Moreover, the side-entrance gives solvent immediate access to residues such as S114 (β4/β5 loop), Y117 (β4/β5 loop), D123 (β5) and H151 (β8) (Fig. 3e), which are the coordinating residues for water molecules in the distal β-barrel lumen in the xlσ1R$_{prog-co}$ structure (Fig. 2b). This data suggests that the side-entrance may be a potential water entrance that allows water to reach the distal region of the β-barrel lumen. Interestingly, Pascarella and colleagues used MD simulations to reveal a potential opening between the β4/β5 loop and the α4 helix (and the β10 strand) of σ1R, which overlaps with the side-entrance, and proposed it for potential ligand access[29]. Therefore, it will require further investigation to dissect the function of the side-entrance of σ1R.

### Structures of xlσ1R bound to a putative agonist DHEAS
In the meantime, we also determined a structure of xlσ1R bound to a putative agonist DHEAS by soaking DHEAS into xlσ1R crystals. The xlσ1R-DHEAS complex structure was solved to 2.50 Å in the *I*432 space group (termed xlσ1R$_{DHEAS-I432}$; Table 1 and Supplementary Fig. 1). The

xlσ1R$_{DHEAS-I432}$ structure contains a single protomer in the asymmetric unit, similar to the xlσ1R$_{prog-co}$ structure. Structural alignment between xlσ1R$_{DHEAS-I432}$ and xlσ1R$_{prog-co}$ revealed that the protein portion of the two structures are nearly identical with an all-atom RMSD of 0.17 Å (Supplementary Fig. 7a). In the β-barrel lumen of xlσ1R$_{DHEAS-I432}$, a clear electron density was observed, which was readily fit with a DHEAS molecule (Fig. 4a–c). Interestingly, superposition of the lumen-lining residues between xlσ1R$_{prog-co}$ and xlσ1R$_{DHEAS-I432}$ yielded an all-atom RMSD of 0.21 Å (Supplementary Fig. 7b). This observation suggests that binding of different neurosteroid ligands (e.g. progesterone vs. DHEAS) may only cause very subtle conformational change in binding site residues (Supplementary Fig. 7b), similar to the comparison between xlσ1R$_{prog-co}$ and xlσ1R$_{unknown-lig}$ (Supplementary Fig. 2e).

The DHEAS molecule binds to a location similar to where progesterone binds (Fig. 4d and Supplementary Fig. 7c), also with its β-face facing the membrane, as the electron density showed clearly two protrusions corresponding to the C10 and C13 methyl groups on DHEAS' β-face (Fig. 4e). Interestingly, unlike progesterone, DHEAS binds with the C17 carbonyl group (on D-ring) near the membrane and the C3 sulfuric ester group (on A-ring) pointing towards the distal end

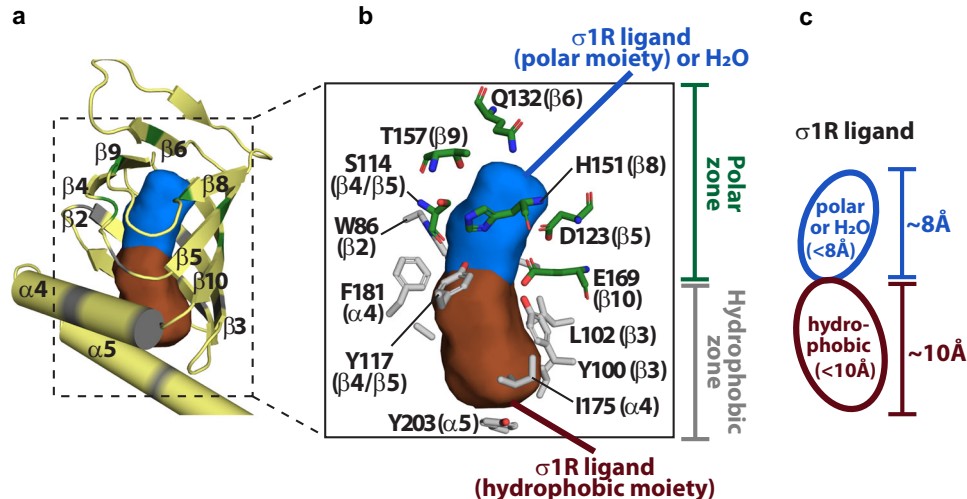

**Fig. 5 | A two-part-interaction model for ligand binding in σ1R. a** A σ1R ligand (shown as surface) is usually elongated in shape and binds inside the β-barrel lumen of xlσ1R (yellow cartoon). Two parts of the ligand are colored in blue and brown, respectively. Residues that interact with the ligand are colored in green or gray. For a clearer view, only the β-barrel and α4/α5 of xlσ1R are shown. **b** A close-up view of panel **a**, showing details of xlσ1R-ligand interactions. Participating residues are shown as sticks. Green residues line the distal region of the β-barrel lumen (polar zone) and interact with the ligand's polar moiety (or water; shown as blue surface). Gray residues line the proximal region of the β-barrel lumen (hydrophobic zone) and interact with the hydrophobic moiety of the ligand (shown as brown surface). Residue Y117 may participate in both polar and hydrophobic interactions, but is displayed as gray sticks in this panel. The side chain of E169 in xlσ1R (or E172 in hσ1R) is located near the junction between the polar and hydrophobic zones. **c** A generalized two-part-interaction model for σ1R ligands. The polar moiety (or water) and the hydrophobic moiety of the ligand are shown as a blue oval and a brown oval, respectively. The maximal lengths of the polar moiety and the hydrophobic moiety of a ligand are estimated to be ~8 Å and ~10 Å, respectively.

of the β-barrel lumen, which is opposite to progesterone (Fig. 4d and Supplementary Fig. 7c). The C17 carbonyl group of DHEAS does not form any hydrogen bond with xlσ1R. However, DHEAS seems to interact directly with the E169 side chain of xlσ1R through its C3 sulfuric ester group, possibly by forming hydrogen bonds (Fig. 4d). Indeed, p$K_a$ of the E169 side chain increases from ~5.5 to ~6.5 after DHEAS binding, as calculated by PROPKA 3.1[36]. This result indicates that, under the crystallization condition of pH 5.5, a considerable portion of the E169 side chain may be protonated, which is capable of forming hydrogen bonds with the C3 sulfuric ester oxygens of DHEAS (Fig. 4d). Consistently, the xlσ1R-E169A mutant displayed a much weaker binding affinity ($K_d$ = 456.2 ± 17.0 μM) for DHEAS compared to the wild-type xlσ1R ($K_d$ = 18.0 ± 2.5 μM; Fig. 4f and Supplementary Table 3), suggesting that E169 plays an important role in DHEAS binding in xlσ1R. Interestingly, as seen in xlσ1R$_{prog-co}$, electron densities of water were also observed in the distal β-barrel lumen of the xlσ1R$_{DHEAS-I432}$ structure (Fig. 4c, g). Two modeled water molecules may form hydrogen bonds with the C3 sulfuric ester oxygens of DHEAS and the D123 (β5) side chain (Fig. 4g), which may help DHEAS binding. Consistently, calculated binding energies between DHEAS and xlσ1R are lower with two water molecules than without water (Supplementary Table 4).

## Discussion

The main finding of this work is the determination of xlσ1R structures bound to progesterone (a putative antagonist) and DHEAS (a putative agonist). It is therefore intriguing to compare binding of these two neurosteroid ligands in xlσ1R, and three common features were observed. First, both progesterone and DHEAS bind to a similar location in the β-barrel lumen of xlσ1R with their β-faces towards the membrane (Fig. 4d and Supplementary Fig. 7c). This orientation may allow an α-π interaction[37] between the steroid's α-face and the Y100 (β3) side chain of xlσ1R (Figs. 1i and 4d). Second, both progesterone and DHEAS bind with the more compact end, i.e. the C3 carbonyl group of progesterone and the C17 carbonyl group of DHEAS, close to the ER membrane (Fig. 4d and Supplementary Fig. 7c). The bulkier end,

i.e. the C17 keto methyl group of progesterone and the C3 sulfuric ester group of DHEAS, reaches towards the distal space of the β-barrel lumen, which is more spacious (Fig. 4d and Supplementary Fig. 7c). Third, in addition to hydrophobic interactions with xlσ1R, both progesterone and DHEAS interact with water molecules in the distal β-barrel lumen (Figs. 2b and 4g), which may stabilize binding of progesterone and DHEAS in xlσ1R (Supplementary Table 4). Meanwhile, prominent differences between the binding of progesterone and DHEAS in xlσ1R were also noted. First, progesterone and DHEAS bind to xlσ1R with opposite directions (Fig. 4d and Supplementary Fig. 7c). That is, progesterone binds to xlσ1R with the steroid A-ring proximal to the membrane while DHEAS with the steroid D-ring near the membrane (Figs. 1f, g and 4b, c). This result indicates that the direction of the steroid rings is not critical during neurosteroid binding in σ1R. Second, in addition to hydrophobic interactions, DHEAS also interacts directly with E169 (β10) of xlσ1R (Fig. 4d), whereas progesterone does not form any direct hydrogen bond with the receptor (Fig. 1i). These data suggest that binding of different neurosteroids in σ1R may employ a common mechanism, but with different binding modes tailored to specific steroid ligands.

Therefore, based on the xlσ1R-progesterone and xlσ1R-DHEAS structures, we propose a generalized two-part-interaction model to describe steroid ligand binding in σ1R (Fig. 5a, b). The first part of interaction occurs in the membrane-proximal region of the β-barrel lumen (the hydrophobic zone; Fig. 5b), which spans ~10 Å from the membrane to E169/E172 (β10) of σ1R (Fig. 5c). This part of interaction is primarily composed of hydrophobic interactions between steroid rings and several hydrophobic residues of β2/β3 strands and α4/α5 helices (Fig. 5b and Supplementary Fig. 3). Meanwhile, the second part of interaction takes place in the membrane-distal region of the β-barrel lumen (the polar zone; Fig. 5b), which measures ~8 Å from E169/E172 (β10) of σ1R to the distal end of the β-barrel lumen (Fig. 5c). This part of interaction consists of mainly polar interactions (hydrogen bonds or salt bridges) between oxygen atoms of the steroids' C3/C17 attachments (or nearby water molecules) and the hydrophilic residues (e.g. E169/E172 of β10) that line the polar zone (Fig. 5b). Notably, Y117 (β4/β5

loop) of xlσ1R is positioned near the interface between the hydrophobic zone and the polar zone, and may participate in both interactions through its phenyl ring and hydroxyl group (Fig. 5b). Supportively, docking models in xlσ1R for seven potential steroid ligands, including progesterone, pregnenolone, pregnenolone sulfate, allopregnanolone, DHEA, DHEAS and 16,17-didehydroprogesterone[29] (Supplementary Table 2), are generally compatible with the two-part-interaction model (Supplementary Fig. 8a-8g). For example, their steroid rings participate mainly in the hydrophobic interactions with β2/β3 strands and α4/α5 helices in the hydrophobic zone. Furthermore, this two-part-interaction model for steroid binding in σ1R is also compatible with its pharmacophore models[7] and the published hσ1R/xlσ1R structures bound to synthetic ligands[8–10]. The major difference for synthetic ligand binding is that a basic nitrogen atom of synthetic ligands forms polar interactions (hydrogen bonds or salt bridges) with E169/E172 (β10) in the polar zone (Supplementary Fig. 8h). Therefore, the two-part-interaction model may potentially be generalized to describe ligand binding in σ1R (Fig. 5c). For instance, one could dock and discuss a ligand in σ1R with the help of the two-part-interaction model. Of note, docking models of progesterone appear consistent with the two-part-interaction model (Supplementary Fig. 8a). However, its orientation and location seems incorrect compared to xlσ1R-progesterone complex structures (Supplementary Fig. 8a). Therefore, analysis of ligand binding in σ1R using the two-part-interaction model needs to be further validated by experimental evidence. Nevertheless, the xlσ1R-progesterone and xlσ1R-DHEAS structures in this study, as well as the two-part-interaction model, may be valuable in generation of a more general pharmacophore model for σ1R ligands.

It is also interesting to discuss the affinity difference between progesterone and DHEAS for xlσ1R. Indeed, DHEAS shows a lower affinity for xlσ1R than progesterone does (Supplementary Table 3), even though progesterone does not have direct polar interactions with xlσ1R while DHEAS interacts directly with the E169 side chain (Figs. 1i and 4d). The affinity difference is consistent with previous functional studies that report progesterone as a more potent σ1R ligand than other neurosteroid ligands (including DHEAS)[12,13]. Also, estimated interaction or binding energies between progesterone and xlσ1R are lower than that between DHEAS and xlσ1R (Supplementary Table 4), indicating a higher affinity for progesterone than DHEAS. In our opinion, in addition to direct hydrogen bonding, other interactions (e.g. hydrophobic interactions and indirect hydrogen bonding) may also contribute to binding affinity of progesterone or DHEAS to σ1R as described in the two-part-interaction model (Fig. 5). For example, progesterone may form more extensive hydrophobic interactions with xlσ1R than DHEAS does. Consistently, compared to DHEAS, progesterone binds closer to the membrane within the β-barrel lumen (Fig. 4d and Supplementary Fig. 7c), which is more hydrophobic than the distal β-barrel lumen region (Fig. 5b). On the other hand, a potential electrostatic repulsion may occur between the sulfuric ester group (negative charge) of DHEAS and the side-chain carboxyl group (negative charge) of E169 (β10), which may destabilize polar interactions of the C3 sulfuric ester group of DHEAS in the distal β-barrel lumen region. Recently, we determined another xlσ1R-DHEAS complex structure to 3.09 Å in a *C*2 form (termed xlσ1R$_{DHEAS-C2}$; Table 1 and Supplementary Fig. 1). The xlσ1R$_{DHEAS-C2}$ structure contains six protomers (two trimers) in an asymmetric unit (Supplementary Fig. 9a). These protomers aligned well between each other and with xlσ1R$_{DHEAS-I432}$ (Supplementary Fig. 9b). Interestingly, DHEAS seems to show three different binding poses in xlσ1R$_{DHEAS-C2}$: Pose-1 for protomers A and D (Supplementary Fig. 9c), Pose-2 for protomers B, E and F (also for xlσ1R$_{DHEAS-I432}$; Supplementary Fig. 9b, d) and Pose-3 for protomer C (Supplementary Fig. 9e). The three DHEAS poses differ mainly in that the C3 sulfuric ester group assumes different conformations (Supplementary Fig. 9c–e). This data may provide potential evidence of flexible (and possibly loose) binding of DHEAS' C3 sulfuric ester group

in the distal β-barrel lumen region of xlσ1R. Meanwhile, due to the resolution limit of xlσ1R$_{DHEAS-C2}$ (3.09 Å), which may not be sufficient to distinguish clearly different conformations of the sulfate group of DHEAS, we interpreted the xlσ1R$_{DHEAS-C2}$ structure with caution and used it only as supplemental evidence to discuss DHEAS binding in xlσ1R.

Another point worth mentioning is the unknown electron density observed in xlσ1R structures without addition of any known ligand, e.g. in xlσ1R$_{unknown-lig}$ (Fig. 1c, d). The unidentified molecule may come from cells, or may be a component of purification or crystallization buffer. It is interesting to note that the shape of the unknown density in xlσ1R$_{unknown-lig}$ (Fig. 1c, d) appears different from those observed in previously reported xlσ1R structures (e.g. PDBs 7W2B and 7W2E). Currently, it is unclear if there are multiple unidentified molecules that could occupy the unknown density in the β-barrel lumen of xlσ1R structures. These issues will require further investigation to address.

## Methods

### Protein expression and purification

The gene encoding wild-type σ1R from *Xenopus laevis* (xlσ1R, NCBI accession NP_001087013.1) was synthesized (Genewiz, China) and cloned into a modified pPICZ plasmid (Thermo Fisher Scientific) containing an amino-terminal tag of decahistidine and tobacco etch virus (TEV) protease recognition site following the hemagglutinin signal peptide. The E169A mutation was introduced by site-directed mutagenesis using QuikChange II system (Agilent) according to manufacturer's recommendation, and was verified by sequencing. The wild-type or mutant xlσ1R was overexpressed in yeast strain GS115 (*Pichia pastoris*) cells by adding 1% (v/v) methanol and 2.5% (v/v) dimethyl sulfoxide (DMSO) at OD$_{600 nm}$ of ~1 and shaking at 20 °C for 48 h. Cell pellets were resuspended in lysis solution (LS) containing 20 mM Tris-HCl pH 7.5, 150 mM NaCl, 10% (v/v) glycerol, 1 mM phenylmethanesulfonyl fluoride (PMSF) and 2 mM β-mercaptoethanol, and lysed by an AH-1500 high-pressure homogenizer (ATS, China) at 1,300 MPa. Undisrupted cells and cell debris were separated by centrifugation at 3000×*g* for 10 min, and membranes were collected by ultracentrifugation at 140,000×*g* for 1 h at 4 °C. Protein was extracted by addition of 1% (w/v) n-dodecyl-β-D-maltopyranoside (DDM, Anatrace) and 0.1% (w/v) cholesteryl hemisuccinate (CHS, Anatrace) at 4 °C for 2 h and the extraction mixture was centrifuged at 200,000 x g for 30 min at 4 °C. The supernatant was then loaded onto a cobalt metal affinity column, washed with 20 bed-volume of LS containing 3 mM DDM, 0.01% (w/v) CHS and 20 mM imidazole pH 8.0, and eluted with LS supplemented with 3 mM DDM, 0.01% (w/v) CHS and 250 mM imidazole pH 8.0.

### Crystallization

Affinity-purified xlσ1R was treated with TEV protease at a 1:20 ratio (TEV: xlσ1R, w/w) for 30 min at 20 °C to remove the purification tag. The protein was then concentrated to 6–8 mg/ml and loaded onto a Superdex 200 Increase 10/300 GL column (Cytiva) equilibrated in 20 mM sodium HEPES pH 7.5, 150 mM NaCl, 5 mM β-mercaptoethanol, 40 mM octyl-β-D-glucopyranoside (OG) and 0.001% (w/v) CHS and was further purified by size-exclusion chromatography (SEC). SEC-purified xlσ1R was then concentrated to 5–6 mg/ml, and 500 nl of protein solution was mixed with an equal volume of crystallization solution manually in a vapor diffusion sitting-drop setup and was incubated at 20 °C. (1) The xlσ1R$_{unknown-lig}$ crystals grew in 0.32 M LiCl, 0.1 M sodium citrate pH 5.5, 12% (w/v) PEG 4000, 10% (v/v) glycerol, and 15 mM sodium cholate hydrate. (2) The xlσ1R$_{prog-soak}$ crystals grew in 0.32 M LiCl, 0.1 M sodium citrate pH 5.5, 12% (w/v) PEG 4000, 10% (v/v) glycerol, 15 mM sodium cholate hydrate, and were soaked with 1 mM progesterone. Progesterone was solubilized in methanol at 100 mM, which was further diluted with the SEC buffer to 10 mM as a stock solution for the current study. (3) The xlσ1R$_{prog-co}$ crystals grew in 0.32 M LiCl, 0.1 M sodium citrate pH 5.5, 12% (w/v) PEG 4000, 10% (v/v)

glycerol, 15 mM sodium cholate hydrate, and 1 mM progesterone. (4) The xlσ1R$_{side-open}$ and xlσ1R$_{side-open-all}$ crystals grew in the same condition as the xlσ1R$_{unknown-lig}$ crystals, and were randomly harvested for data collection. (5) Both xlσ1R$_{DHEAS-C2}$ and xlσ1R$_{DHEAS-I432}$ crystals grew in 0.32 M LiCl, 0.1 M sodium citrate pH 5.5, 12% (w/v) PEG 4000, 10% (v/v) glycerol, 15 mM sodium cholate hydrate, and were soaked with 1 mM DHEAS. The xlσ1R crystals usually appear in 2–3 days, and reach full-size in a week. The crystals were cryo-protected by raising the glycerol concentration to 16% with a 2% (v/v) incremental step, and flash-frozen in liquid nitrogen.

### Data collection, structure solution, and structural analysis
Diffraction data were collected on beamlines BL18U1 and BL19U1[38] of National Facility for Protein Science in Shanghai (NFPS) at Shanghai Synchrotron Radiation Facility (SSRF). The data were indexed, integrated and scaled using the autoPROC pipeline package v1.0.5 (Global Phasing Limited)[39], which includes XDS (BUILT 20230630)[40] and AIMLESS (CCP4 package v7.0.072)[41]. All xlσ1R structures were solved by molecular replacement with Phaser v2.8.1[42] using the published xlσ1R$_{closed-endo}$ structure (PDB entry 7W2B) as a template. Manual model building and refinement was carried out using Coot v0.8.9.1[43] and phenix.refine[44], and Molprobity[45] was used to monitor and improve protein geometry. For xlσ1R$_{side-open}$, xlσ1R$_{side-open-all}$ and xlσ1R$_{DHEAS-C2}$, non-crystallographic symmetry (NCS) was applied during the refinement to improve the map, and was relaxed in the last a few rounds of refinement. For xlσ1R$_{prog-co}$ and xlσ1R$_{prog-soak}$, some discrete positive electron densities in $F_o$-$F_c$ maps were not modeled since the shape of these densities does not match with the shape of buffer or mother liquor components, and these densities are mostly far away (>4 Å) from the xlσ1R protein density. The data collection and refinement statistics were generated using phenix.table_one[44] and the values are listed in Table 1. Electron density maps for all xlσ1R structures and omit maps for all ligand-bound xlσ1R structures generated in this study are shown in Supplementary Fig. 1. All structural figures, RMSD calculations and length measurements were performed in PyMOL v1.8.0.6 (Schrödinger, LLC). Accessibility analysis was performed using the volume-filling program HOLLOW v1.1[46] with default settings. The p$K_a$ values of the E169 side chain of xlσ1R structures were calculated by PROPKA v3.1[36].

### Interaction/binding energy calculation
The interaction energies between xlσ1R and bound progesterone or DHEAS in xlσ1R structures were calculated using the AMMOS2 web server[35] (http://drugmod.rpbs.univ-paris-diderot.fr/ammosHome.php). For each xlσ1R-progesterone or xlσ1R-DHEAS complex structure, the input files for the receptor (xlσ1R, as a.pdb file) and the ligand (progesterone or DHEAS, as a.mol2 file) were prepared according to the AMMOS2 user guide. The binding free energies between xlσ1R and progesterone or DHEAS were computed by BIOVIA Discovery Studio 2021 (Dassault Systèmes). Briefly, the CHARMM[47] force field was applied to the receptor, and in situ ligand energy minimization was conducted before computation of binding free energies. The "Input Atomic Radii" parameter was defined using van der Waals radii, and binding energies were computed using the "Calculate Binding Energies" function. For xlσ1R$_{prog-co}$ and xlσ1R$_{DHEAS-I432}$, the water molecules within the distal space of the β-barrel lumen were either kept or discarded to estimate the contribution of these water molecules to progesterone/DHEAS binding in xlσ1R. Calculated interaction or binding energies were listed in Supplementary Table 4. Coordinates of optimized ligand models of progesterone (Supplementary Data 1-2) or DHEAS (Supplementary Data 3-4) from binding energy calculations were provided as Supplementary Data files.

### Molecular dynamics simulation
MD simulations for the xlσ1R$_{prog-co}$ structure were carried out in GROMACS (BUILT 2023.2)[48] using the CHARMM36m force field[49].

To reduce the computational complexity, the amino-terminal transmembrane helix of xlσ1R$_{prog-co}$ (residues 1-33) and the ligand (progesterone) were removed and one xlσ1R$_{prog-co}$ monomer was prepared as the input model using CHARMM-GUI[50]. The resulting xlσ1R$_{prog-co}$ model was solvated by the 150 mM KCl solution with TIP3P water model[51]. The final simulation box (8.2 nm × 8.2 nm × 8.2 nm) contains a total of 51,745 atoms, including xlσ1R$_{prog-co}$ (residues 34-219), 16,247 water molecules, 52 K⁺ and 46 Cl⁻. The system was first energy minimized using the steepest descent algorithm over 100 steps, and was relaxed by applying restraints using the standard CHARMM-GUI equilibration protocol. Then, the water molecules and ions of the system were equilibrated for 125 ps using an NVT ensemble (constant Number of particles, Volume, and Temperature), followed by a 125-ps NPT equilibration (constant Number of particles, Pressure, and Temperature), while the protein and crystallographic water molecules were fixed. The system was well-equilibrated as indicated by reaching stable temperature, pressure and density over time before production runs. The simulation production run was performed in three parallel runs for 100 ns (with different initial velocities generated by random seeds) without positional restraints at 1-fs steps at a temperature of 303 K and a constant pressure of 1 bar, and RMSD of the protein backbone from its initial to final states was utilized to analyze convergence of simulations. UCSF ChimeraX v1.4[52] was used to visualize the simulation trajectories (1 snapshot per ns for 100 ns) and to export structural coordinates. A simulation checklist is included in Supplementary Table 5. The initial input coordinate file (Supplementary Data 5) and final output coordinate files (Supplementary Data 6–8) of MD simulations were provided as Supplementary Data files.

### Molecular docking
Docking analysis was performed using AutoDock Vina v1.1.2[53] for modeling seven steroids (Supplementary Table 2) into the protein portion of the xlσ1R$_{prog-co}$ structure. The receptor was prepared according to AutoDock Vina manual, and polar hydrogens were added using AutoDock Tools[54]. Coordinates of the steroids were generated by UCSF Chimera v1.15[55] from their SMILES strings, and were prepared according to AutoDock Vina manual by merging non-polar hydrogens and verifying rotatable bonds in AutoDock Tools. The docking grid was set to encompass the β-barrel lumen of xlσ1R$_{prog-co}$ with the aid of model visualization in AutoDock Tools, and docking trials were performed with high exhaustiveness. Docking models with the highest score for each steroid were used for analysis in this study, and their coordinates were provided as Supplementary Data files (Supplementary Data 9-15).

### Microscale thermophoresis
Binding of progesterone or DHEAS to xlσ1R was analyzed by microscale thermophoresis (MST)[56]. MST analysis was performed using Monolith NT.115 (NanoTemper, Germany) by staining His-tagged xlσ1R with the RED-Tris-NTA 2nd Generation dye (NanoTemper, Germany). Affinity-purified xlσ1R wild-type or E169A protein was further purified by SEC in a buffer containing 20 mM sodium HEPES pH 7.5, 150 mM NaCl and 1 mM DDM. Peak fractions were pooled and diluted to 200 nM using the SEC buffer. Protein samples were mixed with 100 nM RED-Tris-NTA 2nd Generation dye at a 1:1 ratio and incubated for 30 min at room temperature. Then the sample was centrifuged at 15,000×$g$ for 10 min at 4 °C to keep the supernatant containing the labeled protein. Progesterone was solubilized in methanol at 100 mM, which was further diluted with the SEC buffer to 10 mM as the stock solution to prepare a diluted series of the titrant, and the highest concentration of the progesterone solutions was 200 μM. Meanwhile, DHEAS was solubilized in the SEC buffer to 7 mM as the stock solution, and the highest concentration of the DHEAS solution series was 7 mM. Labeled xlσ1R was mixed with serial-diluted progesterone or DHEAS

and incubated for 30 min at room temperature. Then the samples were loaded into capillaries and MST measurements were performed using MO.Control v1.6.1 software (NanoTemper, Germany) according to the Monolith manual. Fluorescence signals of ligand binding were normalized in a 0-100% scale. The equilibrium dissociation constant ($K_d$) was determined using the MO.Affinity Analysis v2.2.4 software (NanoTemper, Germany) with the "$K_d$ fit" function. All MST measurements were performed in three biologically independent experiments ($N = 3$). The $K_d$ values are listed in Supplementary Table 3, and are expressed as mean ± SD in the text. Two-tailed Student's t-test was performed using Microsoft Excel for Mac 2016 for statistical analysis in Supplementary Table 3.

### Reporting summary

Further information on research design is available in the Nature Portfolio Reporting Summary linked to this article.

## Data availability

The atomic coordinates and structure factors of the xlσ1R structures generated in this study have been deposited in the Protein Data Bank under the following accession codes: 8W4B (xlσ1R$_{prog-co}$), 8W4C (xlσ1R$_{prog-soak}$), 8W4D (xlσ1R$_{unknown-lig}$), 8WWB (xlσ1R$_{DHEAS-I432}$), 8WUE (xlσ1R$_{DHEAS-C2}$), 8W4E (xlσ1R$_{side-open}$), and 8YBB (xlσ1R$_{side-open-all}$). The MST data and interaction/binding energy values are provided in the Supplementary Information or Source Data file. Coordinates of optimized ligand models of progesterone or DHEAS from binding energy calculations, the initial input coordinate file and final output coordinate files of MD simulations, and docking models were provided as Supplementary Data files. Previously reported hσ1R and xlσ1R structures used in this study are available in the Protein Data Bank under the following accession codes: 5HK1 (hσ1R: PD144418), 5HK2 (hσ1R: 4-IBP), 6DJZ (hσ1R: haloperidol), 6DK0 (hσ1R: NE-100), 6DK1 (hσ1R: pentazocine), 7W2B (xlσ1R$_{closed-endo}$), 7W2C (xlσ1R$_{closed-PRE084}$), 7W2D (xlσ1R$_{closed-SIRA}$), and 7W2E (xlσ1R$_{open-endo}$) Source data are provided with this paper.

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

## Acknowledgements

Diffraction data used in this study were collected on beamlines BL18U1 and BL19U1 of National Facility for Protein Science in Shanghai (NFPS) at Shanghai Synchrotron Radiation Facility (SSRF). The authors thank the staff from these beamlines for assistance during data collection. This work was supported in part by Sichuan Science and Technology Program grant 2023ZYD0125 to X.Z., the National Natural Science Foundation of China (NSFC) grants 31770783 to X.Z., West China Hospital of Sichuan University "1.3.5 Project" for Disciplines of Excellence grants ZYYC20014 to X.Z., and Sichuan University "From 0 To 1" Innovation Program grant 2023SCUH0067 to X.Z.

## Author contributions

X.Z. and Z.S. conceived the project and wrote the manuscript. C.F. and Y.X. performed the structural and functional studies; C.F. and X.Z. performed the computational studies; all authors analyzed the data.

## Competing interests

The authors declare no competing interests.
