## [Peer Review File · Nature Communications]

Insight into binding of endogenous neurosteroid ligands to the sigma-1 receptorREVIEWER COMMENTS

Reviewer #1 (Remarks to the Author):

In this paper, Fu et al. discussed how the sigma-1 receptor (σ 1R) binds with neurosteroids like progesterone and DHEAS, detailing the structural differences in their binding mechanisms. The study determined the crystal structures of the σ 1R in complex with several neurosteroids, along with binding assays and molecular dynamics simulations, to provide insights into the interaction. The authors found that while both progesterone and DHEAS bind to the same location in σ 1R, their binding orientations and interactive patterns with the receptor differ significantly. This research helps the understanding of σ 1R's role in neurosteroid binding, which could have implications for therapeutic strategies targeting neurodegenerative disorders and cancer. While the paper aligns with the publication's standards in terms of scientific novelty and relevance, addressing the outlined concerns is crucial for reinforcing its scientific rigor and ensuring a comprehensive and accurate representation of the research findings.

Major concerns and suggestions:

1. For agonist DHEAS binding, the authors presented three poses that vary mostly in the conformation of the sulfate group. Pose-2 appears to be the dominant conformation, as observed in the 2.5 Å resolution χ l σ 1RDHEAS-I432 structure and three of the six protomers of the 3.1 angstrom χ l σ 1RDHEAS-C2 structure, while pose-1 and pose-3 are observed in the other three protomers of the C2 structure. At 3.1 angstrom resolution, it's more challenging and less reliable to interpret the electron density, especially since the binding site doesn't have strong interactions to the sulfate group. It's not convincing enough the two minor poses are not artifacts of poor-resolution data. Could the authors provide a more robust analysis or evidence to support the existence of these minor poses?
2. The discovery of side-open structures in random C2 crystals is intriguing. Did any χ l σ 1RDHEAS-C2 crystals show the same open conformation? Are residues Y117 and W118 near the crystal packing interface? Is there any evidence of the open-close motion in the higher-resolution structures? (determine the high-resolution I432 structure in a subgroup, e.g.)
3. Some large positive densities observed in Fo-Fc maps for structures 8W4B and 8W4C need clarification or potential assignment of small molecules. Also, the distortion in the DHEAS ligand (ZWY) in structures 8WWB and 8WUE suggests a need for a revised 3D model.

Minor concerns and suggestions:

1. Figure 1b Annotation: Enhancing Figure 1b with clear annotations of χ l σ 1Rprog-co would aid in better understanding.
2. Affinity Explanations for Progesterone and DHEAS: The paper discusses decreased affinity due to a lack of direct interaction with progesterone, yet DHEAS shows direct interaction. Clarification on why the affinity between σ 1R and DHEAS remains in the μ M range despite this would be valuable.
3. Stabilization by Hydrogen Bonds: The paper claims hydrogen bonds with water stabilize the σ 1R-DHEAS interaction, but Table S2 shows no difference in interaction energy with and without water. An explanation for this observation would be helpful.

Reviewer #2 (Remarks to the Author):

Summary overview and evaluation:

The paper by Chunting Fu et al addresses important questions regarding the chaperone Sigma-1 receptor (S1R) "active site" positioning of two known eukaryotic endogenous neurosteroid S1R regulators, progesterone (antagonist) and dehydroepiandrosterone sulfate (DHEAS, agonist). The current work is, in part, a continuation of the S1R ligand binding characteristics of the *Xenopus leavis* S1R (xS1R) from this group. The xS1R, expressed in and purified from yeast, shares strong sequence homologies with other eukaryotic S1Rs including the human S1R (hS1R) and its crystal structure. The basic conclusions regarding the characteristics of the two neurosteroids bound in the xS1R crystal structure are overall well supported;

- (1) progesterone interacts uniquely (i.e. without direct contact with E169) in the beta barrel of the xS1R via a series of six inter coordinated water molecules initiated by the steroid D ring 17- keto methyl group. Three possible entrances for water into the xS1R beta barrel were considered based on a combination of crystallographic and molecular dynamics approaches utilizing co

xS1R/progesterone protomer complexes. These are the entrance between the alpha 4/5 helices, the nearby "hinge entrance", and the "side entrance" near the alpha 4 helix/beta 3 residues. The "side entrance" that interfaces the luminal/membrane surface appears likely to be the most operative for water entrance (and perhaps progesterone entrance as well) and agrees generally with 16,17 didehydroprogesterone S1R access based on in silico studies including MD simulations that predict unique protomer B behaviour for neurosteroid binding from Pascarella et al, IJMS, 3, 2023. The possibility that different protomers of the S1R are not coequal in neurosteroid interactions is further supported by crystallographic evidence in the Fu et al paper

2) DHEAS appears to interact in the beta barrel with the opposite pose to progesterone. DHEAS positions are driven primarily by the steroid A ring 17-sulfate ester oxygen interactions with E169. Three different binding poses for DHEAS in the binding site of a six protomer/two trimer/asymmetric (protomers A,B vs B,E,F vs C) unit were observed by soaking the neurosteroid into xS1R crystals implying flexibility of the protomers for DHEAS (agonist) binding. Does the cocrystal also show the same result? The work is exhaustive in its scope, generally methodologically sound and reasonably interpreted and illustrated. Some concerns remain.

Primary Concerns:

1) The "elephant in the room" is the unknown electron density to which the authors refer with regards to shape and water access. It is far from clear what the unknown electron density is or what its possible functional role(s) might be. The authors may have discovered a fundamental property of the S1R binding site, assuming that xS1R is a valid model of the S1R. What is it? This needs more definition. The unknown ligand appears to be relatively high affinity since it survives the purification and crystallization procedures? Is it unique to the manner by which the xS1R is prepared (ie via yeast vs Sf9 vs prokaryotic cell expression systems)? Is it constitutive and does it also occur in the hS1R or other eukaryotic S1Rs? What attempts have been made to remove the unknown ligand? A resolution of these concerns may be important since the affinities or the poses of the neurosteroids (or other S1R ligands), as presented, may be affected by the combined presence of the unknown density?

2) Please extend the MD beyond 10 ns (100-500 ns), if possible, since a surface contour of S1R protomers from the crystal structures appears to show multiple possible open water access areas of the S1R.

3) Once again no dramatic conformational changes in the S1R structure that could presage functional properties that differentiate antagonists from agonists also occur with the binding of progesterone or DHEAS (or other S1R ligands). Is CryoEM a realistic solution?

Specific Concerns:

1) line 20: "including" should be removed since only progesterone and DHEAS are the utilized in the current work. Other neurosteroids may have unique binding features. Comment also applies to line 84. How does DHEA with only the 17-oxo group bind? Both DHEAS and DHEA are important since DHEAS does not have CNS access and DHEA does.

2) Lines 140 and 264: progesterone is a 17-keto methyl steroid not a C17 acetyl. The detailed chemical structures of progesterone and DHEAS (and DHEA) need to be shown. The structures as shown in Fig. 1e and Fig. 4b lack detail except for the ring designations.

3) Line 156: M90 not referenced in Fig1i.

4) There are likely to be important structural and functional roles of the S1R underwritten in part by known cholesterol interactions and perhaps other bilayer lipids. Please discuss.

5) Fig. S1 seems somewhat superfluous. A table with the PDB files would be sufficient for the interested reader.

Reviewer #3 (Remarks to the Author):

The manuscript reports the determination, by X-ray crystallography, of several structures of the sigma-1 receptor (S1R) by *Xenopus laevis* (XIS1R), two of which are in complex with the S1R agonist dehydroepiandrosterone sulphate (DHEAS) and two with the progesterone antagonist. In addition, ligand binding to both XIS1R and a single residue mutant was experimentally evaluated to investigate the importance of specific S1R-ligand interactions, and molecular dynamics simulations were performed to help identify a putative entry pathway for the water molecules observed in some of the structures.

Noteworthy results.

The structures reported in this work allowed the mode of interaction of S1R with two endogenous neurosteroid compounds to be revealed at high resolution. Both compounds were found to be located within the internal cavity enclosed within the β -barrel domain of S1R, like all synthetic

compounds whose structures have been previously determined in complex with S1R, and to bind to a similar location but with different orientations.

These results are of high significance for the research community working on S1R, for the following reasons. First, while the physiological S1R ligand has not been identified yet, this work reports, for the first time, structures of S1R in complex with endogenous compounds that have demonstrated to bind S1R *in vitro* and modulate its activity *in vivo* and are, therefore, candidate physiological S1R ligands. Second, steroid-based compounds have peculiar structural features (i.e., absence of the basic amine site and rigid condensed-rings scaffold) that are not present in compounds whose crystal structures have been previously determined in complex with S1R. As a consequence, the interactions of S1R with these compounds present important differences with respect to those with synthetic ligands, like the replacement of the basic amine of synthetic compounds with sulphate oxygen or water molecules, in the case of DHEAS and progesterone-bound S1R, respectively, as interaction partners for the conserved E169 of XIS1R (E172 of HsS1R). Additionally, S1R is implicated in a variety of pathological conditions, including neurodegenerative disorders and cancer; accordingly, many synthetic S1R ligands have been demonstrated to exert pharmacological effect in those pathologies and some of them are undergoing clinical trials. For these reasons, the results of this work are of relevance for the wide research community working on one or more of these diseases.

From a methodological point of view, the Authors managed to significantly improve the structure quality. The resolution of the XIS1R structures studied in this work was between 2.15 and 3.09 Å, whereas of those studied in the previous work was in the range 2.85-3.80 Å.

All the analyses have been performed in great detail, and the presentation of the results is very clear and exhaustive. The results have been provided with sufficient context and consideration of previous work and references to previous literature are appropriate. Conclusions and claims are supported by the results and there are no flaws in data analysis, interpretation and conclusions.

Main suggestions to improve the manuscript.

1) The two endogenous neurosteroids studied in this work show similarities and differences in their mode of binding to S1R, but it is not clear to what extent the information about their mode of interaction with S1R can be generalized to other endogenous neurosteroids found to bind S1R *in vitro* and/or activate it *in vivo*, such as those mentioned in the manuscript (lines 60-61: pregnenolone, pregnenolone sulphate, allopregnenolone, dehydroepiandrosterone); 16,17-didehydroprogesterone, which is a human endogenous steroid compound predicted by computational procedures to bind S1R with high affinity (see Ref. 29 in the manuscript); or other endogenous steroids. A comparative analysis of the chemical structures of other selected endogenous steroids with those of progesterone and DHEAS, as well as of other molecules whose structures in complex with S1R has been determined, and a description of the interactions that they might or not establish with S1R based on this analysis would add significant value to the results of this work.

2) In my opinion, the contents of the Discussion section are not entirely appropriate. I would expect this section to comprise comments on all the main findings of the work, and briefly summarize them, if required. Conversely, the whole Discussion section is dedicated to a single, specific, however important, open question about S1R, i.e., how ligand binding is transmitted to the external environment. Accordingly, I would not start a Discussion section with "Finally, ..." as if it was the continuation of a concept expressed before, but I would start it as a new section altogether.

Other suggestions.

I would explicitly state that the findings of this work should be taken into account in the generation of novel and more general pharmacophoric models, e.g., by incorporating the presence of an oxygen, in addition to a nitrogen, as hydrogen bond counterpart of the conserved E172 of human S1R (HsS1R) or E169 of XIS1R, or comprising water molecules for the interaction with the hydrophilic patch described at lines 168-169.

It would be interesting for the reader to know whether the "hinge-entrance" (line 193) and the "side-entrance" (line 204) overlap, at least partially, with sites that have been previously proposed as putative ligand entrances based on structure analyses or MD simulations or are, conversely, completely different from them; I suggest stating this explicitly.

The Authors report that the overall structures of the "prog-co" and "unknown-lig" protomers are

highly similar, with all-atoms RMSD of 0.19 Å; but it would be interesting to calculate the RMSD values of residues lining the S1R binding site, or lumen, to try and detect whether small differences occur in this region: S1R structures determined so far are so highly conserved that even minor differences at the side-chain level might provide clues on ligand binding;

Similarly, it would be interesting to calculate and report the RMSD values of residues lining the S1R binding site, or lumen, between XIS1R "prog-co" and "DHEAS-C2" protomer structures, in addition to the values reported for the overall protomers (lines 297-298)

I would specify whether S1R does or does not comprise any polar group, either from side-chains or main-chain, within hydrogen bond distance from the C3 carbonyl group of progesterone or the C17 carbonyl group of DHEAS in any structure.

Style and language.

The style is generally very clear. However, the grammar might be improved following reading by a native English speaker. A few examples are reported below:

- line 112: "folding" should be "fold"
- line 118: "more unambiguous": better "less ambiguous" or "clearer"
- line 177: "To more quantitatively assess" should be "to provide a quantitative estimate"
- line 206: "the side-entrance is relatively stable that it may be captured" -> "the side-entrance is stable enough to be captured";
- lines 265-266: "which is more spacious to accommodate bulky structures" -> "which is stable enough to accommodate bulky structures" or "which is more spacious, therefore it can accommodate bulky structures".

Reviewer #4 (Remarks to the Author):

The study carried out by Fu et al. resolved and characterized six new relatively high-resolution sigma1R crystal structures, including three bound with progesterone and two with DHEAS. Compared to the synthetic ligands bound with the previous sigma1R structures, these two endogenous neurosteroid ligands, progesterone and DHEAS, were found to have distinct binding poses and characteristics by this work, including the involvement of water molecules and flexible poses. The findings are of significant advancement for the field to understand the molecular mechanism of sigma1R. The study has been well designed and executed, and the manuscript was generally well written.

The following points should be considered to further improve the work:

- i) No convince evidence has been shown or cited that progesterone and DHEAS are specifically antagonist and agonist of sigma1R, respectively, especially at the in vitro level. The authors may want to lower the tone from this perspective, which may actually alleviate their burden in identifying the relevant different mechanistic details for which they could not find, e.g., in the discussion.
- ii) The identifications of the "side entrance" of the water and correspondingly the "side-open" structure are interesting. I assume this is a general pathway that goes beyond just progesterone binding. An extended discussion or analysis in the context of other sigma1R structures and ligands will be informative. Can the ligands also go through this pathway?
- iii) The MD simulation is overly short (just 10 ns). Prolonged simulations of multiple trajectories, and some quantitative analysis will strengthen the argument.

Other minor points:

- 1) Whether progesterone and DHEA are also neurosteroids of xenopus laevis should be discussed. If not, a sequence alignment for all the mentioned residues between human and xenopus laevis should be provided in SI.
- 2) Fig. S2c, the transmembrane helix cannot be that tilted in the membrane, and thereby the ER membrane cannot be that thin, or vice versa.
- 3) Line 654 on page 32, I assume that the authors meant "three snapshots" but not "three trajectories".

Reviewer #1:

In this paper, Fu et al. discussed how the sigma-1 receptor (σ 1R) binds with neurosteroids like progesterone and DHEAS, detailing the structural differences in their binding mechanisms. The study determined the crystal structures of the σ 1R in complex with several neurosteroids, along with binding assays and molecular dynamics simulations, to provide insights into the interaction. The authors found that while both progesterone and DHEAS bind to the same location in σ 1R, their binding orientations and interactive patterns with the receptor differ significantly. This research helps the understanding of σ 1R's role in neurosteroid binding, which could have implications for therapeutic strategies targeting neurodegenerative disorders and cancer. While the paper aligns with the publication's standards in terms of scientific novelty and relevance, addressing the outlined concerns is crucial for reinforcing its scientific rigor and ensuring a comprehensive and accurate representation of the research findings.

Major concerns and suggestions:

1. For agonist DHEAS binding, the authors presented three poses that vary mostly in the conformation of the sulfate group. Pose-2 appears to be the dominant conformation, as observed in the 2.5 Å resolution $\text{x}\sigma$ 1R_{DHEAS-I432} structure and three of the six protomers of the 3.1 angstrom $\text{x}\sigma$ 1R_{DHEAS-C2} structure, while pose-1 and pose-3 are observed in the other three protomers of the C2 structure. At 3.1 angstrom resolution, it's more challenging and less reliable to interpret the electron density, especially since the binding site doesn't have strong interactions to the sulfate group. It's not convincing enough the two minor poses are not artifacts of poor-resolution data. Could the authors provide a more robust analysis or evidence to support the existence of these minor poses?

Response:

Thanks for your comment and question. Structural model building relies on the electron density map, and we totally agree that a low-resolution density map may cause uncertainty in model building. In the original manuscript, three poses of DHEAS have been modeled to best-fit the electron densities in $\text{x}\sigma$ 1R_{DHEAS-C2} (3.1 Å, PDB 8WUE). Meanwhile, we also realize that such a resolution (3.1 Å) may not be sufficient to clearly distinguish different conformations of the sulfate group of DHEAS. Therefore, we determined the $\text{x}\sigma$ 1R_{DHEAS-I432} structure at a higher resolution (2.5 Å, PDB 8WWB), which contains a Pose-2 DHEAS. During revision of the manuscript, we carried out additional $\text{x}\sigma$ 1R crystallization and DHEAS soaking, attempting to

solve more x σ 1R-DHEAS structures at higher resolutions to clearly reveal the minor poses of DHEAS. Unfortunately, this effort was not successful, with most data in 3.2-3.8 Å range. Therefore, in the revised manuscript, we focus on the x σ 1R_{DHEAS-I432} structure (2.5 Å, Pose-2, PDB 8WWB) to describe DHEAS binding in the Results section (Fig. 4 and S6). In this way, the major findings and conclusions of the original manuscript are not changed. To discuss DHEAS poses, the x σ 1R_{DHEAS-C2} structure (3.1 Å, PDB 8WUE) was moved to the Discussion section (Fig. S8), and the tone of DHEAS poses was lowered to only discuss potential binding flexibility of DHEAS in x σ 1R.

2. The discovery of side-open structures in random C2 crystals is intriguing. Did any x σ 1R_{DHEAS-C2} crystals show the same open conformation? Are residues Y117 and W118 near the crystal packing interface? Is there any evidence of the open-close motion in the higher-resolution structures? (determine the high-resolution I432 structure in a subgroup, e.g.)

Response:

Thanks for your insightful comment. As suggested, we have additionally tested diffraction of ~60 x σ 1R crystals soaked with DHEAS recently. However, these crystals diffracted to 3.2-3.8 Å range only. From our experience, DHEAS soaking is quite challenging as it usually leads to worse diffraction of x σ 1R crystals, and at low resolutions it becomes difficult to tell if DHEAS is soaked in or not. As a result, so far, we have only solved three x σ 1R-DHEAS complex structures: two for x σ 1R_{DHEAS-I432} (2.5 Å and 2.9 Å) and one for x σ 1R_{DHEAS-C2} (3.1 Å). The 2.5 Å x σ 1R_{DHEAS-I432} (PDB 8WWB) and the 3.1 Å x σ 1R_{DHEAS-C2} (PDB 8WUE) structures have been reported in the manuscript. Unfortunately, we did not capture a similar side-open conformation in these three x σ 1R-DHEAS structures.

Y117 and W118 are located near the trimer interface between two adjacent protomers within a trimer (Fig. S5a). In x σ 1R_{side-open} (PDB 8W4E), W118 of a side-open protomer (e.g. protomer C) is 4-5 Å away from Q191 and F193 (α 4/ α 5 loop) of protomer A (Fig. S5b). Meanwhile, Y117 or W118 of a side-closed protomer (e.g. protomer A) is not in proximity to its adjacent protomer B (Fig. S5c). Therefore, there is sufficient space at the trimer interface for conformational change of Y117 and W118 between the side-closed and side-open conformations. This description has been added in the Results section.

As suggested, we have also attempted to solve higher-resolution side-open structures of x σ 1R by collecting additional diffraction data (~30 sets) recently. Unfortunately, no higher-resolution

side-open xl σ 1R structure was obtained in this effort. However, a xl σ 1R structure in C2 space group with all six protomers in the side-open conformation (termed xl σ 1R_{side-open-all}) was determined at 3.1 Å (PDB 8YBB) and has been added in the revised manuscript (Fig. S5d-S5f). Now, two C2 xl σ 1R structures with side-open protomers (PDBs 8W4E and 8YBB) are reported in the revised manuscript, showing four and six side-open protomers, respectively. These data suggest that the conformational change of Y117 and W118 between the side-closed and side-open conformations is dynamic during crystallization. This description has been added in the Results section.

3. Some large positive densities observed in F_o-F_c maps for structures 8W4B and 8W4C need clarification or potential assignment of small molecules. Also, the distortion in the DHEAS ligand (ZWY) in structures 8WWB and 8WUE suggests a need for a revised 3D model.

Response:

Thanks for your suggestion. We are also aware of some discrete, unmodeled positive densities in F_o-F_c maps of structures 8W4B and 8W4C. These densities are not modeled mainly due to two reasons. First, the shape of these densities does not match with the shape of buffer or mother liquor components in the crystallization well, making it difficult to assign small molecules to these densities. Second, these densities are mostly far away (>4 Å) from the xl σ 1R protein density, suggesting that they are less likely water molecules and probably would not significantly affect interpretation of the xl σ 1R structure if not modeled. This clarification has been added in the Methods section.

Meanwhile, thanks for pointing out the DHEAS (ZWY) distortion issue. This issue was likely caused by phenix.elbow when it generated ligand restraints for DHEAS from its SMILES string, and several restraints deviate from the ideal bond lengths and angles of DHEAS (ZWY) for some reason. We apologize for not having spotted this issue, and have now fixed the restraint file by manually correcting the wrong bond lengths and angles using phenix.reel. Both models of xl σ 1R_{DHEAS-I432} (PDB 8WWB) and xl σ 1R_{DHEAS-C2} (PDB 8WUE) have been revised using corrected DHEAS restraints, and both PDB entries have been updated with new coordinates as suggested. Updated validation reports for PDB entries 8WWB and 8WUE are provided with this revision.

Minor concerns and suggestions:

1. Figure 1b Annotation: Enhancing Figure 1b with clear annotations of $\text{x}\sigma\text{1R}_{\text{prog-co}}$ would aid in better understanding.

Response:

Thanks for your suggestion. We have re-made Fig. 1b to enhance $\text{x}\sigma\text{1R}_{\text{prog-co}}$ for better presentation as suggested, including annotations of $\alpha\text{1-}\alpha\text{5}$ helices and $\beta\text{1-}\beta\text{10}$ strands.

2. Affinity Explanations for Progesterone and DHEAS: The paper discusses decreased affinity due to a lack of direct interaction with progesterone, yet DHEAS shows direct interaction. Clarification on why the affinity between σ1R and DHEAS remains in the μM range despite this would be valuable.

Response:

Thanks for your comment and suggestion. Indeed, it is interesting to note the affinity difference between progesterone and DHEAS for $\text{x}\sigma\text{1R}$. DHEAS shows a lower affinity for $\text{x}\sigma\text{1R}$ than progesterone does (Table S3), even though progesterone does not have direct polar interactions with $\text{x}\sigma\text{1R}$ while DHEAS interacts directly with the E169 side chain (Fig. 1i and 4d). The affinity difference is consistent with previous functional studies that report progesterone as a more potent σ1R ligand than other neurosteroid ligands (including DHEAS) (Refs 12 and 13). Also, estimated interaction or binding energies between progesterone and $\text{x}\sigma\text{1R}$ are lower than that between DHEAS and $\text{x}\sigma\text{1R}$ (Table S4), indicating a higher affinity for progesterone than DHEAS. In our opinion, in addition to direct hydrogen bonding, other interactions (e.g. hydrophobic interactions and indirect hydrogen bonding) may also contribute to binding affinity of progesterone or DHEAS to σ1R as described in the two-part-interaction model (Fig. 5) (please refer to our response to the first comment of Reviewer #3). For example, progesterone may form more extensive hydrophobic interactions with $\text{x}\sigma\text{1R}$ than DHEAS does. Consistently, compared to DHEAS, progesterone binds closer to the membrane within the β -barrel lumen (Fig. 4d and S6c), which is more hydrophobic than the distal β -barrel lumen region (Fig. 5b). On the other hand, a potential electrostatic repulsion may occur between the sulfuric ester group (negative charge) of DHEAS and the side-chain carboxyl group (negative charge) of E169 (β10), which may destabilize polar interactions of the C3 sulfuric ester group of DHEAS in the distal β -barrel lumen region. The $\text{x}\sigma\text{1R}_{\text{DHEAS-C2}}$ structure with three DHEAS binding poses may provide potential evidence of flexible (and possibly loose) interaction of DHEAS' C3 sulfuric

ester group with E169/E172 of α 1R (Fig. S8). These factors may be the underlying reason for the affinity difference between progesterone and DHEAS, and this description has been added in the Discussion section.

3. Stabilization by Hydrogen Bonds: The paper claims hydrogen bonds with water stabilize the α 1R-DHEAS interaction, but Table S2 shows no difference in interaction energy with and without water. An explanation for this observation would be helpful.

Response:

Thanks for your comment and suggestion. In our opinion, calculated interaction or binding energies may provide supplemental evidence during protein-ligand interaction analysis, and we agree that one single calculation program may not be sufficiently accurate or reliable. For example, in the original Table S2, calculated interaction energy for α 1R_{DHEAS-I432} with 2 water molecules was only slightly lower than without water (-30.9 kcal/mol vs. -30.7 kcal/mol). Therefore, in the revised manuscript, we have further added binding free energy calculations by another program (BIOVIA Discovery Studio) for α 1R_{prog-co} with or without 6 water molecules, and for α 1R_{DHEAS-I432} with or without 2 water molecules. The calculated energies from AMMOS2 and BIOVIA Discovery Studio are largely consistent in that these water molecules lowered interaction or binding energies between progesterone/DHEAS and α 1R. For example, the binding energy between DHEAS and α 1R calculated by BIOVIA Discovery Studio decreased from -43.5 kcal/mol (without water) to -45.6 kcal/mol (with 2 water). This description has been added in the Results section, and binding energies calculated by BIOVIA Discovery Studio have been added in Table S4.

Reviewer #2:

Summary overview and evaluation:

The paper by Chunting Fu et al addresses important questions regarding the chaperone Sigma-1 receptor (S1R) “active site” positioning of two known eukaryotic endogenous neurosteroid S1R regulators, progesterone (antagonist) and dehydroepiandrosterone sulfate (DHEAS, agonist). The current work is, in part, a continuation of the S1R ligand binding characteristics of the *Xenopus leavis* S1R (xS1R) from this group. The xS1R, expressed in and purified from yeast, shares strong sequence homologies with other eukaryotic S1Rs including the human S1R (hS1R) and its crystal structure. The basic conclusions regarding the characteristics of the two neurosteroids bound in the xS1R crystal structure are overall well supported;

(1) progesterone interacts uniquely (i.e. without direct contact with E169) in the beta barrel of the xS1R via a series of six inter coordinated water molecules initiated by the steroid D ring 17-keto methyl group. Three possible entrances for water into the xS1R beta barrel were considered based on a combination of crystallographic and molecular dynamics approaches utilizing co xS1R/progesterone protomer complexes. These are the entrance between the alpha 4/5 helices, the nearby “hinge entrance”, and the “side entrance” near the alpha 4 helix/beta 3 residues. The “side entrance” that interfaces the luminal/membrane surface appears likely to be the most operative for water entrance (and perhaps progesterone entrance as well) and agrees generally with 16,17 didehydroprogesterone S1R access based on in silico studies including MD simulations that predict unique protomer B behaviour for neurosteroid binding from Pascarella et al, IJMS, 3, 2023. The possibility that different protomers of the S1R are not coequal in neurosteroid interactions is further supported by crystallographic evidence in the Fu et al paper

2) DHEAS appears to interact in the beta barrel with the opposite pose to progesterone. DHEAS positions are driven primarily by the steroid A ring 17-sulfate ester oxygen interactions with E169. Three different binding poses for DHEAS in the binding site of a six protomer/two trimer/asymmetric (protomers A,B vs B,E,F vs C) unit were observed by soaking the neurosteroid into xS1R crystals implying flexibility of the protomers for DHEAS (agonist) binding.

Does the cocrystal also show the same result? The work is exhaustive in its scope, generally methodologically sound and reasonably interpreted and illustrated. Some concerns remain.

Response:

Thanks for your insightful comment and question. Unfortunately, although both soaking and co-

crystallization were utilized to produce $\text{x}\sigma\text{1R}$ -DHEAS complex crystals, only the soaking method yielded diffraction-quality crystals for structure solution of $\text{x}\sigma\text{1R}$ -DHEAS complex.

Primary Concerns:

1) The “elephant in the room” is the unknown electron density to which the authors refer with regards to shape and water access. It is far from clear what the unknown electron density is or what its possible functional role(s) might be. The authors may have discovered a fundamental property of the S1R binding site, assuming that xS1R is a valid model of the S1R. What is it? This needs more definition. The unknown ligand appears to be relatively high affinity since it survives the purification and crystallization procedures? Is it unique to the manner by which the xS1R is prepared (ie via yeast vs Sf9 vs prokaryotic cell expression systems)? Is it constitutive and does it also occur in the hS1R or other eukaryotic S1Rs? What attempts have been made to remove the unknown ligand? A resolution of these concerns may be important since the affinities or the poses of the neurosteroids (or other S1R ligands), as presented, may be affected by the combined presence of the unknown density?

Response:

Thanks for your comment and question. We totally agree that the unknown electron density (e.g. in $\text{x}\sigma\text{1R}_{\text{unknown-lig}}$) is an important issue when studying ligand binding in $\text{x}\sigma\text{1R}$. The unidentified molecule may come from cells, or may be a component of purification or crystallization buffer. Since our first observation of the unknown density in $\text{x}\sigma\text{1R}$ structures (Ref 10), we have made continuous efforts to push higher the resolution of $\text{x}\sigma\text{1R}$ crystals, trying to reveal its identity by obtaining a clearer shape of it. Very recently, we determined a $\text{x}\sigma\text{1R}$ structure (no known ligand added) to 1.94 Å (termed $\text{x}\sigma\text{1R}_{\text{OG}}$) (Fig. S9). Similar to $\text{x}\sigma\text{1R}_{\text{unknown-lig}}$, $\text{x}\sigma\text{1R}_{\text{OG}}$ contains an electron density within the β -barrel, with a ring-like head and a long-chain tail that stretches through the $\alpha\text{4}/\alpha\text{5}$ -entrance (Fig. S9a and S9b). At 1.94 Å resolution, the shape of this electron density matches well with octyl- β -D-glucopyranoside (OG/ β -OG) (Fig. S9c and S9d), which is the detergent used during $\text{x}\sigma\text{1R}$ crystallization at ~40 mM. Meanwhile, $\text{x}\sigma\text{1R}$ showed a K_d of 8.3 ± 2.7 mM for OG by MST binding assay (Table S3). It is possible that OG may enter and bind to the ligand binding site of $\text{x}\sigma\text{1R}$ during crystallization in our experimental setup. Therefore, an OG molecule was modeled into the electron density in the β -barrel lumen of $\text{x}\sigma\text{1R}_{\text{OG}}$ (Fig. S9d). However, it is currently unknown if OG binding has any biological meaning in $\text{x}\sigma\text{1R}$ function or not, and this question may be further investigated in the

lab in the future.

On the other hand, since the shape of unknown electron densities in previously reported $\text{x}\sigma\text{1R}$ structures (e.g. PDBs 7W2B and 7W2E, Ref 10) does not seem to match with OG, it is unclear if there are other unidentified molecules that could occupy the unknown density in the β -barrel lumen of $\text{x}\sigma\text{1R}$ structures. Meanwhile, since all $\text{h}\sigma\text{1R}$ structures reported so far contain synthetic ligands (Refs 8 and 9), it is not clear if $\text{h}\sigma\text{1R}$ would contain a similar unknown molecule in the β -barrel if no known ligand is added during purification and crystallization. These issues will require further investigation to address.

In the meantime, purified $\text{x}\sigma\text{1R}$ (in detergent DDM) has also been subject to extensive dialysis in an attempt to remove any unknown ligand bound to $\text{x}\sigma\text{1R}$ during cell culture or purification. Unfortunately, dialyzed $\text{x}\sigma\text{1R}$ sample became less stable as the protein aggregated during gel filtration, and did not yield meaningful MST binding data or diffraction-quality crystals. Therefore, we have added the $\text{x}\sigma\text{1R}_{\text{OG}}$ structure (Fig. S9) and above description in the Discussion section to carefully discuss the unknown electron density observed in those $\text{x}\sigma\text{1R}$ structures.

2) Please extend the MD beyond 10 ns (100-500 ns), if possible, since a surface contour of S1R protomers from the crystal structures appears to show multiple possible open water access areas of the S1R.

Response:

Thanks for your comment and suggestion. We have extended the MD simulation to 100 ns in 3 parallel runs as suggested. The primary goal of the MD is to find potential entrances for water, and we have observed similar openings (e.g. the side-entrance) in the 100-ns simulations (Fig. S4). The description about MD simulations has been updated in the revised manuscript.

3) Once again no dramatic conformational changes in the S1R structure that could presage functional properties that differentiate antagonists from agonists also occur with the binding of progesterone or DHEAS (or other S1R ligands). Is CryoEM a realistic solution?

Response:

Thanks for your comment and suggestion. We totally agree that cryo-EM may be another powerful technique to investigate conformational changes of σ1R during different ligand binding.

We have done some preliminary negative stain EM for $\alpha 1R$ samples, which show fairly uniform and non-aggregated particles (Fig. R1). Cryo-EM study on $\sigma 1R$ is currently undergoing in the lab, and hopefully we could report new findings in the future.

Figure R1. Negative stain EM of a $\alpha 1R$ sample. Image acquired with a JEOL JEM-1400Flash Electron Microscope.

Specific Concerns:

1) line 20: “including” should be removed since only progesterone and DHEAS are the utilized in the current work. Other neurosteroids may have unique binding features. Comment also applies to line 84. How does DHEA with only the 17-oxo group bind? Both DHEAS and DHEA are important since DHEAS does not have CNS access and DHEA does.

Response:

Thanks for your comment. As suggested, text of line 20 and line 84 has been modified to describe progesterone and DHEAS more specifically, and not to make any statement referring to other ligands that are not investigated in the current study. Meanwhile, it is very interesting to consider binding of DHEAS and DHEA in $\sigma 1R$. However, currently we do not have a complex structure for $\alpha 1R$ -DHEA yet. A docking model of DHEA put the ligand deeper into the distal region of β -barrel lumen, but with the same binding orientation as DHEAS in the $\alpha 1R$ -DHEAS complex structures (Fig. S7e and S7f, please see details in our response to the first comment of Reviewer #3). However, as we discussed in that response, it will require further investigation and experimental evidence to address this issue.

2) Lines 140 and 264: progesterone is a 17-keto methyl steroid not a C17 acetyl. The detailed chemical structures of progesterone and DHEAS (and DHEA) need to be shown. The structures as shown in Fig. 1e and Fig. 4b lack detail except for the ring designations.

Response:

Thanks for your comment. As suggested, text of line 140 and line 264 has been modified to describe progesterone as a 17-keto methyl steroid. Detailed chemical structures of progesterone and DHEAS (and other five steroids including DHEA) have been shown in Table S2 in the revised manuscript.

3) Line 156: M90 not referenced in Fig1i.

Response:

Thanks for your comment. Fig. 1i has been updated with M90 annotation as suggested.

4) There are likely to be important structural and functional roles of the S1R underwritten in part by known cholesterol interactions and perhaps other bilayer lipids. Please discuss.

Response:

Thanks for your suggestion. Cholesterol has been shown to interact with the transmembrane helix ($\alpha 1$) of $\sigma 1R$ (Ref 30), which may regulate the orientation of $\alpha 1$ and play a role in $\sigma 1R$ oligomerization. Unfortunately, no cholesterol (or cholesteryl hemisuccinate) molecule has co-crystallized with $h\sigma 1R$ or $xl\sigma 1R$ to date, even though cholesterol (or cholesteryl hemisuccinate) was present during $h\sigma 1R/xl\sigma 1R$ crystallization (Refs 8-10). Therefore, it will be interesting to pursue a $\sigma 1R$ -cholesterol complex structure in future studies, which may help elucidate the role of cholesterol on $\alpha 1$ conformation and $\sigma 1R$ oligomerization. This description has been added in the Results section.

5) Fig. S1 seems somewhat superfluous. A table with the PDB files would be sufficient for the interested reader.

Response:

Thanks for your suggestion. The original Fig. S1 has been replaced with Table S1 in the revised manuscript.

Reviewer #3:

The manuscript reports the determination, by X-ray crystallography, of several structures of the sigma-1 receptor (S1R) by *Xenopus laevis* (XIS1R), two of which are in complex with the S1R agonist dehydroepiandrosterone sulphate (DHEAS) and two with the progesterone antagonist. In addition, ligand binding to both XIS1R and a single residue mutant was experimentally evaluated to investigate the importance of specific S1R-ligand interactions, and molecular dynamics simulations were performed to help identify a putative entry pathway for the water molecules observed in some of the structures.

Noteworthy results.

The structures reported in this work allowed the mode of interaction of S1R with two endogenous neurosteroid compounds to be revealed at high resolution. Both compounds were found to be located within the internal cavity enclosed within the β -barrel domain of S1R, like all synthetic compounds whose structures have been previously determined in complex with S1R, and to bind to a similar location but with different orientations.

These results are of high significance for the research community working on S1R, for the following reasons. First, while the physiological S1R ligand has not been identified yet, this work reports, for the first time, structures of S1R in complex with endogenous compounds that have demonstrated to bind S1R *in vitro* and modulate its activity *in vivo* and are, therefore, candidate physiological S1R ligands. Second, steroid-based compounds have peculiar structural features (i.e., absence of the basic amine site and rigid condensed-rings scaffold) that are not present in compounds whose crystal structures have been previously determined in complex with S1R. As a consequence, the interactions of S1R with these compounds present important differences with respect to those with synthetic ligands, like the replacement of the basic amine of synthetic compounds with sulphate oxygen or water molecules, in the case of DHEAS and progesterone-bound S1R, respectively, as interaction partners for the conserved E169 of XIS1R (E172 of HsS1R). Additionally, S1R is implicated in a variety of pathological conditions, including neurodegenerative disorders and cancer; accordingly, many synthetic S1R ligands have been demonstrated to exert pharmacological effect in those pathologies and some of them are undergoing clinical trials. For these reasons, the results of this work are of relevance for the wide research community working on one or more of these diseases.

From a methodological point of view, the Authors managed to significantly improve the structure

quality. The resolution of the XIS1R structures studied in this work was between 2.15 and 3.09 Å, whereas of those studied in the previous work was in the range 2.85-3.80 Å.

All the analyses have been performed in great detail, and the presentation of the results is very clear and exhaustive. The results have been provided with sufficient context and consideration of previous work and references to previous literature are appropriate. Conclusions and claims are supported by the results and there are no flaws in data analysis, interpretation and conclusions.

Main suggestions to improve the manuscript.

1) The two endogenous neurosteroids studied in this work show similarities and differences in their mode of binding to S1R, but it is not clear to what extent the information about their mode of interaction with S1R can be generalized to other endogenous neurosteroids found to bind S1R in vitro and/or activate it in vivo, such as those mentioned in the manuscript (lines 60-61: pregnenolone, pregnenolone sulphate, allopregnanolone, dehydroepiandrosterone); 16,17-didehydroprogesterone, which is a human endogenous steroid compound predicted by computational procedures to bind S1R with high affinity (see Ref. 29 in the manuscript); or other endogenous steroids. A comparative analysis of the chemical structures of other selected endogenous steroids with those of progesterone and DHEAS, as well as of other molecules whose structures in complex with S1R has been determined, and a description of the interactions that they might or not establish with S1R based on this analysis would add significant value to the results of this work.

Response:

Thanks for your insightful comment and suggestion. As suggested, we have now included seven steroids, including progesterone, pregnenolone, pregnenolone sulfate, allopregnanolone, DHEA, DHEAS and 16,17-didehydroprogesterone (Table S2), to discuss a potential, generalized model for ligand binding (including steroids) in σ 1R. These steroids share a similar steroidal four-ring scaffold with C10 and C13 methyl groups on the β -face, but with different double-bonds in the steroid rings and different attachments on C3 and C17 of the steroid rings (Table S2).

Combining with docking models of these steroids (Fig. S7a-S7g), structures of xl σ 1R-progesterone and xl σ 1R-DHEAS complexes suggest that binding of these steroid ligands may be described in a two-part-interaction model (Fig. 5). The first part of interaction occurs in the

membrane-proximal region of the β -barrel lumen (the hydrophobic zone) (Fig. 5b), which spans ~ 10 Å from the membrane to E169/E172 ($\beta 10$) of $\sigma 1R$ (Fig. 5c). This part of interaction is primarily composed of hydrophobic interactions between steroid rings and several hydrophobic residues of $\beta 2/\beta 3$ strands and $\alpha 4/\alpha 5$ helices (Fig. 5b and S2). Meanwhile, the second part of interaction takes place in the membrane-distal region of the β -barrel lumen (the polar zone) (Fig. 5b), which measures ~ 8 Å from E169/E172 ($\beta 10$) of $\sigma 1R$ to the distal end of the β -barrel lumen (Fig. 5c). This part of interaction consists of mainly polar interactions (hydrogen bonds or salt bridges) between oxygen atoms of the steroids' C3/C17 attachments (or nearby water molecules) and the hydrophilic residues (e.g. E169/E172 of $\beta 10$) that line the polar zone (Fig. 5b).

Furthermore, this two-part-interaction model for steroid binding in $\sigma 1R$ is also compatible with the published h $\sigma 1R$ /xl $\sigma 1R$ structures bound to synthetic ligands (Table S1). The major difference for synthetic ligand binding is that a basic nitrogen atom of synthetic ligands forms polar interactions (hydrogen bonds or salt bridges) with E169/E172 ($\beta 10$) in the polar zone (Fig. S7h). Therefore, the two-part-interaction model may potentially be generalized to describe ligand binding in $\sigma 1R$ (Fig. 5c). For instance, one could dock and discuss a ligand in $\sigma 1R$ with the help of the two-part-interaction model. Of note, docking models of progesterone appear consistent with the two-part-interaction model (Fig. S7a). However, its orientation and location seems incorrect compared to xl $\sigma 1R$ -progesterone complex structures (Fig. S7a). Therefore, analysis of ligand binding in $\sigma 1R$ using the two-part-interaction model needs to be further validated by experimental evidence. This description has been added in the Discussion section.

2) In my opinion, the contents of the Discussion section are not entirely appropriate. I would expect this section to comprise comments on all the main findings of the work, and briefly summarize them, if required. Conversely, the whole Discussion section is dedicated to a single, specific, however important, open question about S1R, i.e., how ligand binding is transmitted to the external environment. Accordingly, I would not start a Discussion section with “Finally, ...” as if it was the continuation of a concept expressed before, but I would start it as a new section altogether.

Response:

Thanks for your comment. The Discussion section has been re-written as suggested. In the revised manuscript, the Discussion section is focused on comparison and discussion of steroid

ligand (and other ligand) binding in σ 1R.

Other suggestions.

1) I would explicitly state that the findings of this work should be taken into account in the generation of novel and more general pharmacophoric models, e.g., by incorporating the presence of an oxygen, in addition to a nitrogen, as hydrogen bond counterpart of the conserved E172 of human S1R (HsS1R) or E169 of X1S1R, or comprising water molecules for the interaction with the hydrophilic patch described at lines 168-169.

Response:

Thanks for your comment and suggestion. As suggested, we have discussed a more generalized two-part-interaction model to describe ligand binding in σ 1R. Please also see details in our response to your first comment. The xl σ 1R-progesterone and xl σ 1R-DHEAS structures in this study, as well as the two-part-interaction model, may be valuable in generation of a more general pharmacophore model for σ 1R ligands. This statement has been added in the Discussion section.

2) It would be interesting for the reader to know whether the “hinge-entrance” (line 193) and the “side-entrance” (line 204) overlap, at least partially, with sites that have been previously proposed as putative ligand entrances based on structure analyses or MD simulations or are, conversely, completely different from them; I suggest stating this explicitly.

Response:

Thanks for your comment and suggestion. The currently available h σ 1R/xl σ 1R structures suggest that the hinge-entrance and the side-entrance do not overlap with the α 4/ α 5-entrance, which has been proposed for ligand entry by our group (Ref 10). Interestingly, Pascarella and colleagues used MD simulations to reveal a potential opening between the β 4/ β 5 loop and the α 4 helix (and the β 10 strand), which overlaps with the side-entrance, and proposed it for potential ligand access (Ref 29). It is possible that σ 1R may contain multiple entry sites for different ligands. Meanwhile, it will also require further investigation, e.g. by capturing a σ 1R structure with a ligand entering through the side-entrance, to thoroughly address this issue. This description has been added in the Results section. Please also refer to our response to the second comment of Reviewer #4.

3) The Authors report that the overall structures of the “prog-co” and “unknown-lig” protomers are highly similar, with all-atoms RMSD of 0.19 Å; but it would be interesting to calculate the RMSD values of residues lining the S1R binding site, or lumen, to try and detect whether small differences occur in this region: S1R structures determined so far are so highly conserved that even minor differences at the side-chain level might provide clues on ligand binding;

Response:

Thanks for your comment. As suggested, we have now aligned the lumen-lining residues between $\text{xl}\sigma\text{1R}_{\text{prog-co}}$ and $\text{xl}\sigma\text{1R}_{\text{unknown-lig}}$ structures, and obtained an all-atom RMSD of 0.22 Å for these residues (Fig. S1e). Indeed, a visual examination of these aligned residues revealed very subtle differences. For example, the hydroxy oxygens of Y117 of the two structures are only 0.4 Å apart (Fig. S1e). Given that the B-factors (ADPs) of these lumen-lining residues of the two structures range from 30 Å² to 50 Å², their thermal equilibrium movement is estimated to be 0.6-0.8 Å. Therefore, from our understanding, the lumen-lining residues of $\text{xl}\sigma\text{1R}_{\text{prog-co}}$ and $\text{xl}\sigma\text{1R}_{\text{unknown-lig}}$ structures do not show a significant difference beyond their thermal equilibrium movement. The description about the RMSD of lumen-lining residues has been added in the Results section.

4) Similarly, it would be interesting to calculate and report the RMSD values of residues lining the S1R binding site, or lumen, between X1S1R “prog-co” and “DHEAS-C2” protomer structures, in addition to the values reported for the overall protomers (lines 297-298).

Response:

Thanks for your comment. In our response to the first comment of Reviewer #1, we have moved the $\text{xl}\sigma\text{1R}_{\text{DHEAS-C2}}$ structure to the Discussion section (Fig. S8), and are focusing on $\text{xl}\sigma\text{1R}_{\text{DHEAS-1432}}$ to describe DHEAS binding in the revised manuscript (Fig. 4 and S6). Therefore, we have superimposed the lumen-lining residues between $\text{xl}\sigma\text{1R}_{\text{prog-co}}$ and $\text{xl}\sigma\text{1R}_{\text{DHEAS-1432}}$ structures, which yielded an all-atom RMSD of 0.21 Å for these residues (Fig. S6b).

5) I would specify whether S1R does or does not comprise any polar group, either from side-chains or main-chain, within hydrogen bond distance from the C3 carbonyl group of progesterone or the C17 carbonyl group of DHEAS in any structure.

Response:

Thanks for your comment. No oxygen, nitrogen or sulfur atom of α 1R is located within 3.8 Å from the oxygen atom of the C3 carbonyl group of progesterone or the C17 carbonyl group of DHEAS in any α 1R structure in this study. Since the distance cutoff for hydrogen bonds is usually 3.5-3.6 Å in practical, there is no direct hydrogen bond (if any, it would be very weak) between α 1R and the C3 carbonyl group of progesterone or the C17 carbonyl group of DHEAS. This description has been added in the Results sections as suggested.

Style and language.

The style is generally very clear. However, the grammar might be improved following reading by a native English speaker. A few examples are reported below:

- line 112: “folding” should be “fold”
- line 118: “more unambiguous”: better “less ambiguous” or “clearer”
- line 177: “To more quantitatively assess” should be “to provide a quantitative estimate”
- line 206: “the side-entrance is relatively stable that it may be captured” -> “the side-entrance is stable enough to be captured”;
- lines 265-266: “which is more spacious to accommodate bulky structures” -> “which is stable enough to accommodate bulky structures” or “which is more spacious, therefore it can accommodate bulky structures”.

Response:

Thanks for your comment and suggestion. We apologize for the grammatic issues and have edited the text as suggested. We have also carefully gone through the manuscript several times to address grammatic issues as much as we could, and will consult an editing service if still necessary.

Reviewer #4:

The study carried out by Fu et al. resolved and characterized six new relatively high-resolution sigma1R crystal structures, including three bound with progesterone and two with DHEAS. Compared to the synthetic ligands bound with the previous sigma1R structures, these two endogenous neurosteroid ligands, progesterone and DHEAS, were found to have distinct binding poses and characteristics by this work, including the involvement of water molecules and flexible poses. The findings are of significant advancement for the field to understand the molecular mechanism of sigma1R. The study has been well designed and executed, and the manuscript was generally well written.

The following points should be considered to further improve the work:

i) No convince evidence has been shown or cited that progesterone and DHEAS are specifically antagonist and agonist of sigma1R, respectively, especially at the in vitro level. The authors may want to lower the tone from this perspective, which may actually alleviate their burden in identifying the relevant different mechanistic details for which they could not find, e.g., in the discussion.

Response:

Thanks for your comment and suggestion. Indeed, classification of progesterone and DHEAS in the literature as “antagonist” and “agonist”, respectively, is primarily based upon their pharmacological effects in vivo. As suggested, we have lowered the tone to describe progesterone and DHEAS as “putative antagonist” and “putative agonist”, respectively. Meanwhile, in our response to the second comment of Reviewer #3, the Discussion section has been re-written to focus on discussion of steroid ligand binding (and other ligand binding) in σ 1R to make this story more coherent. Therefore, conformational difference between agonist vs. antagonist binding to σ 1R would not be discussed in the revised manuscript, and that topic could make another interesting story for future investigation.

ii) The identifications of the “side entrance” of the water and correspondingly the “side-open” structure are interesting. I assume this is a general pathway that goes beyond just progesterone binding. An extended discussion or analysis in the context of other sigma1R structures and ligands will be informative. Can the ligands also go through this pathway?

Response:

Thanks for your insightful comment. Interestingly, Pascarella and colleagues used MD simulations to reveal a potential opening between the $\beta 4/\beta 5$ loop and the $\alpha 4$ helix (and the $\beta 10$ strand), which overlaps with the side-entrance, and proposed it for potential ligand access (Ref 29). This description has been added in the Results section. It is possible that $\sigma 1R$ may contain multiple entry sites for different ligands. The side-entrance observed in the $xl\sigma 1R_{\text{side-open}}$ structure measures $\sim 6.2 \text{ \AA} \times 2.8 \text{ \AA}$, which is too small to pass progesterone ($\sim 7.5 \text{ \AA} \times 5.9 \text{ \AA}$ measured transversely), DHEAS ($\sim 7.6 \text{ \AA} \times 6.2 \text{ \AA}$), PRE-084 ($\sim 7.5 \text{ \AA} \times 5.5 \text{ \AA}$), or S1RA ($\sim 7.6 \text{ \AA} \times 5.2 \text{ \AA}$). However, it is also possible that the $xl\sigma 1R_{\text{side-open}}$ structure may not represent a fully-open state of the side-entrance, which may be then sufficiently large for ligands to go through. It will certainly require further investigation, e.g. by capturing a $\sigma 1R$ structure with a ligand entering through the side-entrance, to thoroughly address this issue.

iii) The MD simulation is overly short (just 10 ns). Prolonged simulations of multiple trajectories, and some quantitative analysis will strengthen the argument.

Response:

Thanks for your suggestion. We have extended the MD simulation to 100 ns in 3 parallel runs for analysis as suggested. The primary goal of the MD is to find potential entrances for water, and we have observed similar openings (e.g. the side-entrance) in the 100-ns simulations (Fig. S4). The description of MD simulations has been updated in the revised manuscript.

Other minor points:

1) Whether progesterone and DHEAS are also neurosteroids of *xenopus laevis* should be discussed. If not, a sequence alignment for all the mentioned residues between human and *xenopus laevis* should be provided in SI.

Response:

Thanks for your comment. We could not find related literatures that specifically address if progesterone and DHEAS are neurosteroids of *Xenopus laevis*. Therefore, a sequence alignment between $h\sigma 1R$ and $xl\sigma 1R$ (and mouse $\sigma 1R$) is provided in Fig. S2 with all the mentioned residues highlighted as suggested.

2) Fig. S2c, the transmembrane helix cannot be that tilted in the membrane, and thereby the ER

membrane cannot be that thin, or vice versa.

Response:

Thanks for your comment. In all $\alpha 1$ structures in this study, two $\alpha 1$ trimers pack in a membrane-side-to-membrane-side manner (Fig. S1a), forcing $\alpha 1$ helices between the two trimers to fold inside toward the trimer bottom. In this way, the $\alpha 1$ trimers pack more tightly than previously reported $\alpha 1$ structures, and this is probably the reason why $\alpha 1$ crystals in this study diffracted better than previous ones. As a result, the $\alpha 1$ helix of all $\alpha 1$ structures in this study is more tilted than previously reported (Fig. 1b and S1c). Since $\alpha 1$ does not participate directly in ligand binding inside the β -barrel lumen, $\alpha 1$ was not included during discussion of neurosteroid ligand binding in $\alpha 1$ in this manuscript. Therefore, the membrane indication in Fig. S2c (now Fig. S1c in the revised manuscript) is illustrated only to reflect $\alpha 1$ being a transmembrane helix. This clarification has been added in the Results section.

3) Line 654 on page 32, I assume that the authors meant “three snapshots” but not “three trajectories”.

Response:

Thanks for your comment and correction. We apologize for the incorrect term usage. The text has been edited accordingly.

REVIEWERS' COMMENTS

Reviewer #1 (Remarks to the Author):

Having thoroughly reviewed the author's response and the revised manuscript, I find that all concerns and comments have been appropriately addressed. The revisions adequately reflect the suggestions made during the review process. Therefore, I fully support the publication of this manuscript by Fu et al in Nature Communications.

Reviewer #2 (Remarks to the Author):

General

The revised paper by Fu et al is improved. The authors have reasonably addressed most of the reviewers' concerns in appropriate detail. Clarifications of results and an expanded discussion section have been added as a result of the reviewers' comments. A focus on 2.5 Å, Pose 2 for DHEAS binding of xS1R 1432, reinforcement of the side-open parameters of protomers A-F and the possibility that the S1R may have multiple ligand entry pathways are important take aways from the manuscript. The authors have now identified limitations of their current data, made appropriate adjustments to interpretations and to their conclusions and suggested several additional directions for relevant xS1R structural/ligand (additional neurosteroids?).

Noteworthy Additional Comments:

Because of the hydrogen bonding network as proposed in the current work for progesterone (less so for DHEAS) and perhaps for other similar neurosteroids (or synthesized ligands), a high affinity or covalent consitutive molecule with hydrogen bonding capabilities could be a problem for accurate interpretation of S1R/ligand structures. This is especially a concern when the S1R is expressed and purified from yeast (or other eukaryotic cell lines) that contain robust steroid-like or other S1R relevant metabolic pathways. Unfortunately, true identification of the unknown "constitutive?" ligand goes unresolved in the current work. The authors offer a xS1R/octyl beta-D glucopyranoside (OG) co-crystal structure (at an exceptional resolution of 1.94 Å) as a possibility for the unkown ligand or its facsimile. OG is unlikely to be the candidate unknown ligand although it may fit the suggested "proximal/distal" idea for hydrophobic/hydrophilic ligands binding to S1R (Fig. 5). That OG is unlikely to be the unknown ligand is supported by the authors own data (e.g Table 1) comments and conclusive statements (e.g lines 414,415). Therefore, references and discussion (i.e. Fig S9, Methods Section and Discussion Section) to the xS1R/OG data are best removed from the MS because they detract from an otherwise effective (although unqualified with regards to the unknown ligand) crystallographic presentations of progesterone and DHEAS binding to the xS1R beta barrel. With a Kd of 8.3 mM (Table S3), a potential competing ligand of Kd 1 micromolar (e.g Prog. in Fig.1j and Table 3) or 10-18 micromolar (DHEAS in Fig.4f and Table 3) or certainly even higher affinity ligands under the crystallization conditions reported should readily displace OG. [As an additional note but with lesser significance (when the OG data is removed from the paper)is that the actual binding curves for OG are not shown. OG has a critical micelle concentration (CMC) of roughly 14 mM in water around "room temperature". It is likely that the OG binding conditions reported under the MST method (7mM-100 mM/lines 527-530) would have resulted in most of the OG as a micelle even when accounting for the possible CMC-reducing ionic effects of HEPES, NaCl,etc. A complete binding curve for OG would have also required concentrations well below 7 mM (lowest conc reported) which is too close to the Kd of 8.3 mM. for an accurate resolution of a binding curve]

Reviewer #3 (Remarks to the Author):

The Authors have thoroughly addressed all of my previous comments.

Veronica Morea

Reviewer #4 (Remarks to the Author):

The authors have addressed all my comments satisfactorily.

REVIEWERS' COMMENTS

Reviewer #1 (Remarks to the Author):

Having thoroughly reviewed the author's response and the revised manuscript, I find that all concerns and comments have been appropriately addressed. The revisions adequately reflect the suggestions made during the review process. Therefore, I fully support the publication of this manuscript by Fu et al in Nature Communications.

Response:

Thanks for your comment.

Reviewer #2 (Remarks to the Author):

General

The revised paper by Fu et al is improved. The authors have reasonably addressed most of the reviewers' concerns in appropriate detail. Clarifications of results and an expanded discussion section have been added as a result of the reviewers' comments. A focus on 2.5 Å, Pose 2 for DHEAS binding of xS1R I432, reinforcement of the side-open parameters of protomers A-F and the possibility that the S1R may have multiple ligand entry pathways are important take aways from the manuscript. The authors have now identified limitations of their current data, made appropriate adjustments to interpretations and to their conclusions and suggested several additional directions for relevant xS1R structural/ligand (additional neurosteroids?).

Response:

Thanks for your comment.

Noteworthy Additional Comments:

Because of the hydrogen bonding network as proposed in the current work for progesterone (less so for DHEAS) and perhaps for other similar neurosteroids (or synthesized ligands), a high affinity or covalent constitutive molecule with hydrogen bonding capabilities could be a problem for accurate interpretation of S1R/ligand structures. This is especially a concern when the S1R is expressed and purified from yeast (or other eukaryotic cell lines) that contain robust steroid-like or other S1R relevant metabolic pathways. Unfortunately, true identification of the unknown "constitutive?" ligand goes unresolved in the current work. The authors offer a xS1R/octyl beta-D glucopyranoside (OG) co-crystal structure (at an exceptional resolution of 1.94 Å) as a possibility for the unknown ligand or its facsimile. OG is unlikely to be the candidate unknown ligand although it may fit the suggested "proximal/distal" idea for hydrophobic/hydrophilic ligands binding to S1R (Fig. 5). That OG is unlikely to be the unknown ligand is supported by the authors own data (e.g Table 1) comments and conclusive statements (e.g lines 414,415).

Therefore, references and discussion (i.e. Fig S9, Methods Section and Discussion Section) to the xS1R/OG data are best removed from the MS because they detract from an otherwise effective (although unqualified with regards to the unknown ligand) crystallographic presentations of progesterone and DHEAS binding to the xS1R beta barrel. With a K_d of 8.3 mM (Table S3), a potential competing ligand of K_d 1 micromolar (e.g Prog. in Fig.1j and Table 3) or 10-18 micromolar (DHEAS in Fig.4f and Table 3) or certainly even higher affinity ligands under the crystallization conditions reported should readily displace OG. [As an additional note but with lesser significance (when the OG data is removed from the paper) is that the actual binding curves for OG are not shown. OG has a critical micelle concentration (CMC) of roughly 14 mM in water around "room temperature". It is likely that the OG binding conditions reported under the MST method (7 mM-100 mM/lines 527-530) would have resulted in most of the OG as a micelle even when accounting for the possible CMC-reducing ionic effects of HEPES, NaCl, etc. A complete binding curve for OG would have also required concentrations well below 7 mM (lowest conc reported) which is too close to the K_d of 8.3 mM for an accurate resolution of a binding curve]

Response:

Thanks for your comment and suggestion. We agree that the OG-related data added very limited information and value to the main findings and conclusions of this work. As suggested, references and discussion to the xS1R-OG data have been removed from the revised manuscript.

Reviewer #3 (Remarks to the Author):

The Authors have thoroughly addressed all of my previous comments.

Response:

Thanks for your comment.

Reviewer #4 (Remarks to the Author):

The authors have addressed all my comments satisfactorily.

Response:

Thanks for your comment.